# Disentangling Transformer Language Models as Superposed Topic Models

**Jia Peng Lim**
Singapore Management University
`jiapeng.lim.2021@smu.edu.sg`

**Hady W. Lauw**
Singapore Management University
`hadywlauw@smu.edu.sg`

## Abstract

Topic Modelling is an established research area where the quality of a given topic is measured using coherence metrics. Often, we infer topics from Neural Topic Models (NTM) by interpreting their decoder weights, consisting of top-activated words projected from individual neurons. Transformer-based Language Models (TLM) similarly consist of decoder weights. However, due to its hypothesised superposition properties, the final logits originating from the residual path are considered uninterpretable. Therefore, we posit that we can interpret TLM as superposed NTM by proposing a novel weight-based, model-agnostic and corpus-agnostic approach to search and disentangle decoder-only TLM, potentially mapping individual neurons to multiple coherent topics. Our results show that it is empirically feasible to disentangle coherent topics from GPT-2 models using the Wikipedia corpus. We validate this approach for GPT-2 models using Zero-Shot Topic Modelling. Finally, we extend the proposed approach to disentangle and analyse LLaMA models.

## 1 Introduction

The term 'superposition' in machine learning typically refers to overlapping concepts. Elhage et al. (2021) allude to the property of superposition in transformers when discussing the difficulty of interpreting its multilayer perceptrons' (MLP) neurons in Transformer Language Models (TLM). Black et al. (2022) also find neurons in transformers having polysemantic behaviour. We believe it is possible to interpret neurons in superposition, motivated by three key insights.

Our first insight lies in the similarity between TLM and Neural Topic Models (NTM) (see Section 2). Many analyses of transformers examine some variation of neuron activations *paths* (Shazeer et al., 2020; Dong et al., 2021). Again, NTM also uses paths (albeit usually across one layer) and interprets the highest activated logits from a given

neuron as a topic. It is thus natural to treat the interpretation of TLM as a topic model with topics in superposition. Arora et al. (2018) show that polysemous words, i.e., multiple meanings, are in linear superposition within word embeddings. Similarly, we posit to linearly disentangle the logit interpretation of neurons in superposition and examine the neurons in the residual stream (see Section 4). We treat neurons from MLP as belonging to the higher-dimensional residual stream.

Our second insight acknowledges the decade-long interpretability research stemming from topic models. Since Blei (2012), there has been a push to reconcile the evaluation of topic models with the human notion of coherence (Mimno et al., 2011; Lau et al., 2014; Röder et al., 2015), and still debated (Doogan and Buntine, 2021; Hoyle et al., 2021). We show that their findings might help advance the interpretation of TLM (see Section 3.1).

Our third insight contributes towards tackling the disentanglement problem from a novel angle. We map the disentanglement problem to an NP-Hard classical graph problem and propose a computationally feasible approach to solve it via heuristics and exact solving (see Section 5). Wary of mirages (Schaeffer et al., 2023), we propose a non-trivial baseline and empirically show that the disentangled superposition interpretations are indeed meaningful and not due to chance (see Section 6).

**Contributions**. This work aims to deploy topic modelling methodologies, specifically in the area of interpretability, to automate the search and evaluation of concepts within TLMs. We propose a novel weights-based, corpus-agnostic approach to disentangle topics from decoder-only TLM with automated evaluation. These extracted topics are constrained to the themes in the chosen corpus. In a static setting, when considering each GPT-2's neurons (residual, MLP) individually, we empirically show that many are in superposition and may embody multiple distinct or related concepts.

TLMs are pre-trained on a large number of tokens across a plethora of themes. One possible application is to mine the TLM to identify and explain the neurons' role(s) concerning these learnt themes. To further validate our results dynamically, we employ a topic modelling task on GPT-2 in a zero-shot manner (ZSTM, see Section 7). Given a corpus as an input, we register top activating neurons and investigate the projected topics. Finally, we extend our proposed approach[1] on LLaMA (see Section 8), detailing our intuition and approach in overcoming the barrier to interpretability introduced by its choice of tokenization.

## 2 Related Work

**Alignment Research.** The area closest to our work is mechanistic interpretability (Olah et al., 2020), a white-box approach to explain models via model weights. Many works focus on explaining attention (Voita et al., 2019; Clark et al., 2019; Hao et al., 2021). Geva et al. (2022) show that sub-updates in feed-forward network layers are interpretable. Millidge and Black (2022) finds interpretable singular value decompositions of Transformer weights. Dar et al. (2023) analyse transformers in the embedding space. Elhage et al. (2022) and Henighan et al. (2023) analyse properties of superposition in toy transformers. More recently, Cunningham et al. (2023) and Bricken et al. (2023) use sparse autoencoders to extract interpretable features from neurons. Without introducing additional neural networks, our work contributes towards the automated discovery and evaluation of superpositions in neurons from TLMs.

**Topic Models.** While earlier works utilised probabilistic graphical models (Blei et al., 2003), recent works focus on neural-based approaches. There are many NTMs (Miao et al., 2016; Srivastava and Sutton, 2017; Zhang and Lauw, 2020; Dieng et al., 2020; Zhang et al., 2022) that incorporate an autoencoder framework (Kingma and Welling, 2014), while others (Yang et al., 2020; Pham and Le, 2021; Zhang et al., 2023) utilizes graph neural networks (Kipf and Welling, 2017; Liu et al., 2019). TLMs have a considerable influence on such neural topic models. Bianchi et al. (2021a), Grootendorst (2022), Han et al. (2023) use TLM embeddings in their topic modelling approaches. Bianchi et al. (2021b) propose an NTM that undergoes self-supervised training on pre-trained multi-

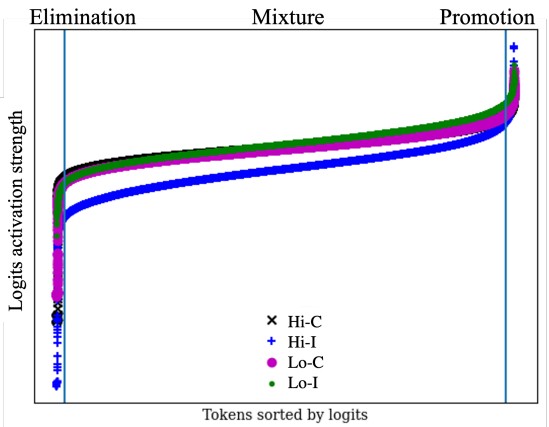

Figure 1: The logit distribution of different $f_n$ in GPT-2-XL with three observable areas: 1) *Elimination* at the left tail, 2) *Mixture* in the middle, and 3) *Promotion* at the right tail. Vertical lines are $\tau$ thresholds to shortlist token pools. Different paths have similar logit distribution.

lingual S-BERT document embeddings (Reimers and Gurevych, 2019) for use in a zero-shot cross-lingual setting. Wang et al. (2023) propose a TLM-agnostic method to add additional concept tokens, where from its representation, inferring topics after fine-tuning to any specific supervised task. For our zero-shot topic modelling task, we do not employ additional training or modifications to GPT-2.

## 3 Disentangling Topics in Superposition

For any given neuron $n$ in language model $M$, different inputs to $M$ may activate the same neuron to express different concepts. We consider these concepts to be in superposition within $n$, and describable in a human-interpretable manner using groups of words – topics. If we can obtain its final logit representation $f_n \in \mathbb{R}^{|S|}$ on token-space $S$, we posit that it is empirically possible to linearly disentangle $f_n$ into multiple interpretable vector representations $i$ in vocabulary space $V$ with a leftover abstract vector representation $a_n$ (Equation 1).

$$f_n = a_n + \sum_{k=1}^{K} i_{k,n}, \forall n \in M \qquad (1)$$

Similar to the behaviours observed in Geva et al. (2022), we observed that $f_n$ is dividable into three distinct behaviours: *promotion*, *elimination*, and *mixture* (Figure 1). If $i_n$ exists, we expect $i_n$ to promote or eliminate specific themes. For uninterpretable *mixture* behaviour, we liken it to $a_n$.

---

[1]github.com/PreferredAI/superposed-topics

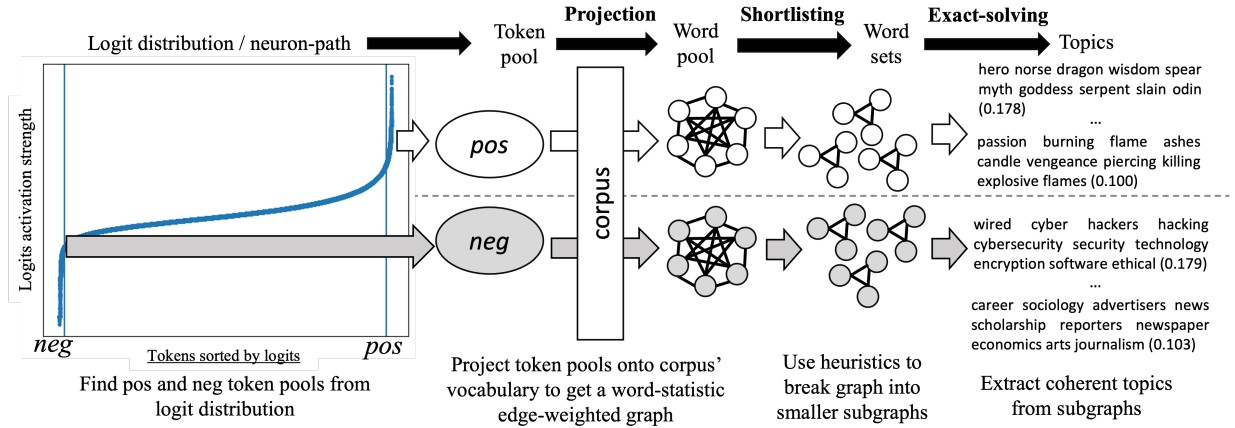

Figure 2: Illustration of our three-step approach. Topic examples are from a random $f^{\text{Lo-I}}$ in GPT-2-XL. The word graphs shown are for illustration purposes. Values in parenthesis are the respective topics' $\text{NPMI}_W$.

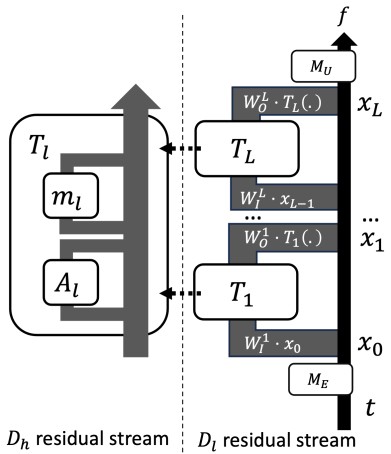

Figure 3: Illustration of a high-level abstraction of the decoder-only transformer model in higher-dimensional $D_h$ and lower-dimensional $D_l$ residual stream.

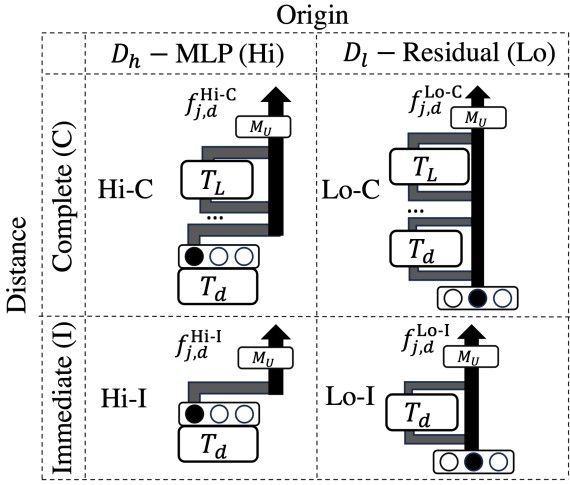

Figure 4: Visualisation of neuron paths $f_{j,d}^{\text{path}}$. Paths (Origin-Distance) projected from individual neurons.

### 3.1 Three-Step Disentanglement Approach

Figure 2 provides an overview of our framework, and we outline the three steps below.

**Projection**. We begin by projecting $f_n$ onto corpus statistics. Intuitively, knowing what to look for helps in the search. We use Wikipedia as our reference corpus and Normalised Pointwise Mutual Information (NPMI) as our corpus statistic. Introduced in Bouma (2009), the NPMI score of a given topic is the mean of NPMI between each possible word-pairs in the group (Equation 2). Lau et al. (2014) and Röder et al. (2015) found correlations between NPMI scores and human ratings on a subset of Wikipedia corpus. We calculate NPMI on Wikipedia's statistics ($\text{NPMI}_W$) and ZSTM's selected corpus statistics ($\text{NPMI}_C$).

$$npmi(w_i, w_j) = \frac{\log \frac{P(w_i, w_j) + \epsilon}{P(w_i) \cdot P(w_j)}}{-\log(P(w_i, w_j) + \epsilon)} \quad (2)$$

We use Wikipedia-EN corpus statistics prepared in Lim and Lauw (2023) (Wikipedia-V40K[2]) that contains its top 40K most frequent vocabulary excluding stop-words. For NPMI calculated from this corpus statistics, Lim and Lauw (2023) benchmarked its correlation to human judgement at $\bar{\rho} = 0.66$, with 40 unique study participants split across eight different study groups. We reiterate that it is possible to use other corpora for this projection step (see Section 7).

---

[2]github.com/PreferredAI/topic-metrics, with article bodies processed using Attardi (2015).

**Shortlisting**. Since our interests lie in $i_n$, the tails of $f_n$, we start by shortlisting the top-$\tau$ and bottom-$\tau$ tokens, respectively termed *pos* and *neg* token pool. Producing word pools from these token pools, we subsequently individually segment into word sets via heuristics using corpus statistics. For some $n$, this step may produce zero word sets, suggesting the concepts within $n$ do not lie within the corpus domain.

**Exact Solving**. Finally, within each word set, we locally optimize it against $\text{NPMI}_W$ to get a $b$-sized coherent topic. A standard fixed topic size allows parity in comparison across differently-sized word sets and ease of human identification. We validate our approach in Section 6, where we show empirically that the distribution of tails $i_n$ differs from a non-trivial random distribution.

## 4  Path Definitions

**Transformer Architecture.** From Elhage et al. (2021), we adopt a similar notation style. However, we further differentiate the residual stream into higher and lower dimensions, respectively $D_h$ and $D_l$, as illustrated in Figure 3. We analyse the activation paths of decoder-only transformers. Given a pre-trained language model $M$ with $L$ layers, with tokens $t$ from token-space $S$, we get its initial embeddings $x_0$ via embedding layer $M_E$ (Equation 3). At each layer $l$, $x_{l-1} \in \mathbb{R}^{D_l}$ increases its dimensions via $W_I^l$, and is passed into Transformer block $T_l$. The $D_h$-output of $T_l$ is then transformed back to $D_l$ space via $W_O^l$ (Equation 3). Final logits $f \in \mathbb{R}^{|S|}$ obtained using unembedding layer $M_U$ on the $D_l$-residual layer $x_L$ (Equation 4).

$$x_l = \begin{cases} M_E(t) & l = 0 \\ T_l(x_{l-1} \cdot W_I^l) \cdot W_O^l & 0 < l \leq L \end{cases} \quad (3)$$

$$f = M_U(x_L) \quad (4)$$

Presenting a higher-level abstraction, we omit details that differ between different models, such as positional embeddings and normalisation. Usually, in each layer $l$, attention is applied via the various attention heads $H_l$ (Equation 5) (Vaswani et al., 2017). Attention layer $A_l$ is connected to MLP $m_l$ in a residual manner (Equation 6).

$$A_l(x) = \sum_{h \in H_l} h(x) \quad (5)$$

$$T_l(x) = x + A_l(x) + m_l(x + A_l(x)) \quad (6)$$

**Different Paths**. There are many possible paths from any $n \in M$ to the final logit layer. We constrain our search to origin-distance pairs with origin (higher (Hi) or lower (Lo) residual dimension) and distance (immediate (I) or complete (C)). Figure 4 visualises the different activation paths for the four modes. Let neuron path $f_{j,d}^{\text{path}}$ represent the final logits of neurons (Equation 7), at index $d$ in layer $j$. Its $D$-dimension boolean representation (Equation 8), with $D_l$ and $D_h$ representing lower and higher dimensions, as the initial input and injected into starting layer $j$ (Equations 9,10,11,12).

$$f_{j,d}^{\text{path}} = M_U(x_{j,d,L}^{\text{path}}) \quad (7)$$

$$n_d = (b_1, \ldots, b_D) : b_d = 1, b_{\neg d} = 0 \quad (8)$$

For Paths Hi-C and Hi-I, we examine MLP neurons in $T_j$ computing all subsequent layers after layer $j$ (Equation 9) with limits $[1, L]$. Hi-C obtains complete neuron path $f_{j,d}^{\text{Hi-C}}$ (Equation 7). However, Hi-I only computes its transformation to the $D_l$-dimension (Equation 10). Starting from $D_h$-MLP is equivalent to evaluating path Lo on layer $j + 1$ on a compositional input.

$$x_{j,d,l}^{\text{Hi-C}} = \begin{cases} n_d \cdot W_O^j & 0 \leq d < D_h \\ & l = j \\ T_l(x_{j,d,l-1}^{\text{Hi-C}} \cdot W_I^l) \cdot W_O^l & j < l \leq L \end{cases} \quad (9)$$

$$x_{j,d,L}^{\text{Hi-I}} = n_d \cdot W_O^j, \ 0 \leq d < D_h \quad (10)$$

For Paths Lo-C and Lo-I, we examine neurons in $D_l$-residual layer $j$ before $T_j$. Path Lo-C computes complete path (Equation 11) and Lo-I only computes layer $j$ (Equation 12).

$$x_{j,d,l}^{\text{Lo-C}} = \begin{cases} n_d & 0 \leq d < D_l \\ & l = j - 1 \\ T_l(x_{j,d,l-1}^{\text{Lo-C}} \cdot W_I^l) \cdot W_O^l & j \leq l \leq L \end{cases} \quad (11)$$

$$x_{j,d,L}^{\text{Lo-I}} = T_j(n_d \cdot W_I^j) \cdot W_O^j, \ 0 \leq d < D_l \quad (12)$$

From each $f_{j,d}^{\text{path}}$, we shortlist top-$\tau$ and bottom-$\tau$ tokens, forming *pos* and *neg* token pools representing $i_n$. After projecting it onto a corpus, we obtain word pools that we represent as word graphs.

## 5  Methodology

In a word pool $G$, each word $v \in V$ is connected via word-pairs $(v_i, v_j) \in E$ weighted by corpus statistic $w_{v_i,v_j}$. We can formulate extracting $b$-sized cliques from $G$ as disentangled topics into the following graph problem.

**Algorithm 1:** Star Heuristic

---

**Function** Star_Heuristic(*graph G, edge limits θ, size limits κ*):

    Delete $(v_i, v_j) \in G$ **if** $w_{v_i,v_j} < \theta_l \vee w_{v_i,v_j} > \theta_u$;

    Delete $v \in G$ **if** $|(v, v_j) \in G| < \kappa_l$;

    Stars ← {Create_Star$(v, G) \forall v \in G$};

    Order star ∈ Stars descending via star.mean_score;

    Chosen ← ∅;

    Sets ← ∅;

    **for** *star* ∈ *Stars* **do**

        **if** *star.centre* ∈ *Chosen* **then**

            continue;

        **end**

        $g ← \{v \in$ star.neighbours$|v \notin$ Chosen$\}$;

        **if** $g \geq \kappa_u$ **then**

            $g ← \{v |$ top $\kappa_u - 1$ largest $w_{\text{star.centre},v} : v \in$ star.neighbours$\}$;

        **end**

        $g ← g\cup$ star.centre;

        **if** $|g| \geq \kappa_l$ **then**

            Sets ← Sets ∪ g;

        **end**

        **for** $v \in g$ **do**

            Chosen ← Chosen ∪ v;

        **end**

    **end**

    **return** *Sets*

**Function** Create_Star($v, G$):

    **return** star.centre ← $v$,

    star.neighbours ← $\{v_j | (v, v_j) \in G\}$,

    star.mean_score ← $\frac{\sum_{v_j \in \text{star.neighbours}} w_{v,v_j}}{|\text{star.neighbours}|}$;

---

$$\max_{i,j} \sum_{i,j \in E} e_{i,j} w_{i,j} \tag{13a}$$

$$\text{s.t} \quad x_i = \{0,1\}, \forall i \in V \tag{13b}$$

$$e_{i,j} = \{0,1\}, \forall i,j \in E \tag{13c}$$

$$\sum_v^V x_v = b \tag{13d}$$

$$\sum_{i,j \in E} e_{i,j} = \frac{b(b-1)}{2} \tag{13e}$$

$$e_{i,j} \leq x_i, \forall i,j \in E \tag{13f}$$

$$e_{i,j} \leq x_j, \forall i,j \in E \tag{13g}$$

## 6 Automated Quantitative Evaluation

**GPT-2** (Radford et al., 2019) is widely used in mechanistic interpretability research. We use model sizes ranging from S (124M) to XL (1558M) (See Table 16 in Appendix B for details). Token space $S_{\text{GPT-2}}$ contains 50,257 tokens encoded using the byte-pair encoding (BPE) algorithm (Sennrich et al., 2016). We map 31K of GPT-2's tokens directly to 19K words ∈ Wikipedia-V40K's $V_{\text{Wiki}}$. No further processing is required, as we recover a large portion of $V_{\text{Wiki}}$.

**Baseline**. We use a random distribution of logits to derive a non-trivial set of coherence topics that occur due to chance. In a random word pool, the number of possible $b$-sized topics scales exponentially with $\tau$, resulting in favourable odds of producing topics that create a false illusion of interpretability. The random baseline uses the same sampling process $S$ and hyper-parameters of GPT-2 to generate 50,000 random token pools.

**Sampling Parameters**. To shortlist the important tails of $f$, we tune hyper-parameters using the first 200 neurons from the first layer of different paths (See Appendix A.1). We locally optimize each word set on NPMI$_W$ to get a topic of size 10.

**Sampling Observations**. Comparing GPT-2 to its random baseline, differences in word sets/neuron are significant (see Table 1). We observe that word pools contain hundreds of words, with *pos* pools double the size of *neg* pools. Additionally, *pos* pools have more word sets per neuron than *neg* pools, although the mean size of word sets remains similar. The random baseline produces more word sets per neuron than *neg* pools but produces fewer word sets than *pos* pools while sampling a significantly larger number of words.

**Maximum Edge-Weighted $b$-Clique (MEWC)**. Given a graph $G$, with vertices $V$ and edges $E$, find max-weighted $b$-sized clique in $G$. This problem is NP-hard (Dijkhuizen and Faigle, 1993), as it reduces to the maximum clique problem (Karp, 1972). As $G$ has a non-trivial amount of vertices, finding a top arbitrary number of non-overlapping MEWC in $G$ is computationally challenging. To enable feasible computation to search for highly-weighted $b$-cliques, we design a *star* heuristic (Algorithm 1), inspired by Park et al. (1996)'s facet-defining star inequalities, breaking $G$ into smaller sub-graphs for exact solving.

**Exact Solving of Subgraphs.** Gouveia and Martins (2015) explored the feasibility of solving different convex formulations of MEKC on $G$ of varying degrees and sizes. We adapt Park et al. (1996)'s formulation to fit our use case. We arrive at a Mixed Integer Program (Eq. 13) that maximises the weight of selected edges (Eq. 13a), selecting edges $(i, j)$ and vertex $i$ by boolean variables $e_{i,j}$ (Eq. 13f) and $x_i$ (Eq. 13g) respectively. We constrain clique size to $b$ (Eq. 13d, 13e) and selection of edges within selected clique (Eq. 13f, 13g).

| modes | Words/Word Pool | | Word Sets/Word Pool | | Words/Word Set | |
|---|---|---|---|---|---|---|
| | *pos* | *neg* | *pos* | *neg* | *pos* | *neg* |
| S-Hi-C | 388.1 ± 122.1 | 216.680 ± 87.9 | 6.90 ± 5.43 | 2.07 ± 1.89 | 15.19 ± 4.84 | 12.82 ± 3.30 |
| S-Hi-I | 480.0 ± 104.9 | 229.119 ± 87.0 | 9.86 ± 5.79 | 2.19 ± 2.22 | 14.37 ± 4.26 | 12.65 ± 3.21 |
| S-Lo-C | 326.4 ± 125.2 | 228.968 ± 86.1 | 5.92 ± 5.00 | 1.70 ± 1.11 | 15.10 ± 4.76 | 12.49 ± 2.91 |
| S-Lo-I | 437.3 ± 113.1 | 336.300 ± 114.1 | 8.11 ± 5.18 | 3.56 ± 2.79 | 14.47 ± 4.38 | 12.74 ± 3.14 |
| XL-Hi-C | 473.0 ± 129.8 | 250.104 ± 83.9 | 11.34 ± 5.29 | 3.08 ± 4.09 | 14.88 ± 4.55 | 13.66 ± 4.00 |
| XL-Hi-I | 478.2 ± 96.0 | 244.775 ± 106.4 | 10.02 ± 3.73 | 3.84 ± 5.10 | 14.60 ± 4.58 | 14.00 ± 4.17 |
| XL-Lo-C | 466.4 ± 102.6 | 244.433 ± 79.7 | 10.02 ± 5.06 | 2.48 ± 3.64 | 14.78 ± 4.51 | 13.44 ± 3.96 |
| XL-Lo-I | 450.4 ± 89.2 | 351.506 ± 106.3 | 6.88 ± 4.10 | 3.99 ± 3.55 | 13.76 ± 3.94 | 13.01 ± 3.42 |
| Random | 559.2 ± 14.2 | | 5.77 ± 1.87 | | 12.04 ± 2.37 | |

Table 1: Sampling statistics of GPT-2. Full results in Table 11 in App. B. Using the Mann-Whitney U test, differences in distributions between GPT-2 and random baseline on number of Word Sets/Word Pool are significant at $p < 0.001$. For GPT-2, there are no empty word pools, i.e. each word pool contains some vocabulary from $V_{\text{Wiki}}$.

| modes | Mean num. Topics | | Top-1 Topic (Topical Pool) $\text{NPMI}_W$ score | | Mean $\text{NPMI}_W$ score per Topical Pool | | Total num. of topics | | % of Abstract Pool | | |
|---|---|---|---|---|---|---|---|---|---|---|---|
| | *pos* | *neg* | *pos* | *neg* | *pos* | *neg* | *pos* | *neg* | *pos* | *neg* | *both* |
| S-Hi-C | 6.76 | 2.08 | 0.202 ± 0.06 | 0.216 ± 0.09 | 0.130 | 0.164 | 211,612 | 22,106 | 7.5 | 35.6 | 6.0 |
| S-Hi-I | 9.76 | 2.19 | 0.218 ± 0.06 | 0.166 ± 0.08 | 0.119 | 0.132 | 351,968 | 24,634 | 1.1 | 34.8 | 0.6 |
| S-Lo-C | 5.68 | 1.70 | 0.215 ± 0.07 | 0.199 ± 0.09 | 0.142 | 0.162 | 40,865 | 4,725 | 11.0 | 34.9 | 7.4 |
| S-Lo-I | 8.10 | 3.56 | 0.227 ± 0.07 | 0.191 ± 0.08 | 0.122 | 0.117 | 70,502 | 21,913 | 2.8 | 16.6 | 1.2 |
| XL-Hi-C | 10.88 | 2.98 | 0.232 ± 0.07 | 0.160 ± 0.07 | 0.127 | 0.126 | 3,135,250 | 291,738 | 3.1 | 34.0 | 0.6 |
| XL-Hi-I | 9.12 | 3.78 | 0.199 ± 0.05 | 0.160 ± 0.06 | 0.117 | 0.118 | 2,665,849 | 535,733 | 2.4 | 26.9 | 0.0 |
| XL-Lo-C | 9.62 | 2.47 | 0.227 ± 0.07 | 0.155 ± 0.07 | 0.128 | 0.126 | 712,337 | 59,817 | 1.8 | 34.2 | 0.9 |
| XL-Lo-I | 6.88 | 4.00 | 0.212 ± 0.06 | 0.182 ± 0.07 | 0.111 | 0.113 | 508,179 | 217,321 | 1.9 | 14.6 | 0.3 |
| Random | 5.77 | | 0.145 ± 0.03 | | 0.064 | | 288,091 | | 0.54 | | |

Table 2: Aggregated results across all layers. Using the Mann-Whitney U test, differences in distributions on top-1 $\text{NPMI}_W$ scores between GPT-2-sampled and random baseline are significant at $p < 0.001$. Topical Pools and Abstract Pools are pools that respectively produce some topics and no topics. We consider a word is frequently occurring (FOW) when it is in more than 2.5% of the topics and remove "stop topics" with more than five FOW.

**Disentangled Topics**. The results in Table 2 show significant differences between random baseline and GPT-2's *pos* pools in top-1 and mean $\text{NPMI}_W$ score. Although there is a close difference between random baseline and *neg* pools in top-1 $\text{NPMI}_W$ score, there is still a sizable difference between their mean $\text{NPMI}_W$ score. Across different model sizes, larger models seem to generate more topics per $f^{\text{path}}$. Additionally, while Top-1 $\text{NPMI}_W$ for *pos* pools remains consistent, we see a slight decline of scores in $\text{NPMI}_W$ in Top-1 $\text{NPMI}_W$ for *neg* pools and mean $\text{NPMI}_W$. Compared to *pos* pools, *neg* pools have more abstract $f^{\text{path}}$ producing zero word sets. From these results, we empirically show that it is possible to disentangle multiple superposed topics.

**Case Study.** We showcase an example from a random $f^{\text{path}}$ in GPT-2-S (Table 3). The extracted topics have similar and varied themes: finance (#1), visuals (#2), places (#3, 4), and sports (#5, 6).

## 7 Zero-Shot Topic Modelling (ZSTM)

In a traditional topic modelling task, we seek to derive $K$ number of topics of size $b$ for corpus analysis. In this ZSTM task, since the pre-training of the TLM is not specific to the selected corpora, the corpus becomes the analysis tool for TLM instead. If our interpretation of its neurons is valid, given a corpus, we expect neurons with similar concepts to be activated frequently. Otherwise, there will be a contradiction. Obtaining a competitive topic set will validate our previous findings. We do not require additional tokens, prompting, or fine-tuning.

**Approach**. We implement a voting scheme giving each token some votes for each layer, examining neurons that exceed the vote threshold (See Appendix A.2). For GPT-2, we extract disentangled topics from Paths Lo-C and Lo-I of the selected neurons. To select the final $K$ topics, we use a greedy algorithm (Lim and Lauw, 2022) optimizing on corpus $\text{NPMI}_C$ with diversity constraints.

| Word pool (alphabetically-sorted) | Word Sets | Extracted Topics (NPMI$_W$) | # |
|---|---|---|---|
| ability adult arch avenue avenues away ball bar bare bay black cap ceiling chance clock commercial competition consumers corner crossed dark demand distribution domestic draw edge est fair financial finely floor free grab green ground hall hand hands held holding ind kneeling level lightly line lines long lower lying market markets minute minutes open pale platform platforms price product resting round run sell shape sit sitting slip spot stage straight street surface thin trade trophy upper value vert waved white wholesale wide wing wood yard (456 ommitted words) | domestic wholesale fair sell price product financial markets commercial value demand consumers trade free market | wholesale price demand sell product value market markets domestic consumers (0.197) | 1 |
| | dark green white vert waved finely thin black spot wing cap long upper edge wide bare lightly surface adult ground shape distribution lower pale | long pale black white thin green upper wide dark finely (0.187) | 2 |
| | arch crossed lying wood clock kneeling resting bar hall bay sit sitting level ceiling corner floor | sit hall lying resting corner sitting ceiling arch floor kneeling (0.106) | 3 |
| | avenue avenues yard street est line lines platforms platform ind | est line lines ind yard platform platforms avenue avenues street (0.103) | 4 |
| | trophy open stage away competition round held minute minutes draw | stage open round draw away trophy minute held minutes competition (0.102) | 5 |
| | holding ability grab straight hand hands slip run chance ball | slip run straight hands hand ability holding chance ball grab (0.069) | 6 |

Table 3: GPT-2-S, Path Hi-C, Layer 11 Neuron 245, *pos* pool. More examples in Appendix C.

| Model | NPMI$_C$ | NPMI$_W$ | TU | $A$ (%) |
|---|---|---|---|---|
| ProdLDA | -0.059 | 0.045 | 0.903 | - |
| ProdLDA-$\beta$ | 0.085 | 0.075 | 0.911 | - |
| CTM | -0.060 | 0.052 | 0.916 | - |
| CTM-$\beta$ | 0.016 | 0.070 | 0.945 | - |
| BERTopic | -0.002 | 0.069 | 0.887 | - |
| GPT-2-S-Lo-C | **0.164** | **0.123** | **0.955** | 20.57 |
| GPT-2-S-Lo-I | **0.169** | **0.115** | **0.950** | 20.57 |
| GPT-2-M-Lo-C | **0.177** | **0.117** | **0.965** | 23.73 |
| GPT-2-M-Lo-I | **0.186** | **0.120** | 0.920 | 23.73 |
| GPT-2-L-Lo-C | **0.197** | **0.115** | 0.945 | 14.99 |
| GPT-2-L-Lo-I | **0.170** | **0.118** | **0.965** | 14.99 |
| GPT-2-XL-Lo-C | **0.184** | **0.112** | 0.940 | 15.37 |
| GPT-2-XL-Lo-I | **0.179** | **0.140** | **0.955** | 15.37 |

(a) 20NewsGroup, $K = 20$. Min. word-pair count = 50.

| Model | NPMI$_C$ | NPMI$_W$ | TU | $A$ (%) |
|---|---|---|---|---|
| ProdLDA | -0.028 | 0.007 | 0.741 | - |
| ProdLDA-$\beta$ | -0.023 | 0.014 | 0.830 | - |
| CTM | 0.027 | 0.050 | 0.895 | - |
| CTM-$\beta$ | 0.067 | 0.057 | 0.835 | - |
| BERTopic | -0.049 | 0.023 | 0.906 | - |
| GPT-2-S-Lo-C | **0.135** | 0.021 | 0.850 | 32.31 |
| GPT-2-S-Lo-I | **0.143** | 0.043 | 0.885 | 32.31 |
| GPT-2-M-Lo-C | **0.156** | 0.021 | 0.865 | 46.52 |
| GPT-2-M-Lo-I | **0.148** | 0.042 | 0.865 | 46.52 |
| GPT-2-L-Lo-C | **0.164** | 0.028 | 0.855 | 38.20 |
| GPT-2-L-Lo-I | **0.143** | 0.033 | 0.855 | 38.20 |
| GPT-2-XL-Lo-C | **0.180** | 0.023 | 0.845 | 43.39 |
| GPT-2-XL-Lo-I | **0.159** | 0.045 | 0.850 | 43.39 |

(b) M10, $K = 20$. Min. word-pair count = 0.

Table 4: A subset of results for topic set sizes $K = 20$. The best baseline scores for the metric are underlined. Bolded PLM scores are better than the best baseline. The complete results are in Appendix B. Full results for topic set sizes $K = 20$ is in Table 14, $K = 50$ is in Table 13. Baseline scores are the mean of 10 independent runs with error bars in Table 15. $A$ is the percentage of neuron paths $f^{\text{path}}$ examined, determined by token votes.

**Corpus**. We generate sets of 20 topics for 20NewsGroup[3], BBC-News (Lim and Buntine, 2014), M10 (Greene and Cunningham, 2006), and DBLP (Tang et al., 2008; Pan et al., 2016) processed from OCTIS (Terragni et al., 2021), with additional sets of 50 topics for 20NewsGroup and BBC-News. Corpus details in Appendix A.2.

**Metrics**. To account for rare occurrences, we set a minimum count of word pairs for NPMI$_C$ calculation, measuring relevance to the corpus. NPMI$_W$ evaluates the generality and coherence of the topic. Desiring uniqueness amongst topics, Topic Uniqueness (TU) (Dieng et al., 2020), measuring the proportion of unique words in $K$ topics. To obtain competitive sets of results, we must overcome a three-way trade-off between the metrics.

**Baselines**. We use three baselines. **ProdLDA** (Srivastava and Sutton, 2017) and **CTM** (Bianchi et al., 2021a) are both popular and competitive autoencoder-based NTM trained on the pre-defined train and validation sets in OCTIS. We derive additional sets of results by selecting the best composite topics (suffixed with -$\beta$) using the same greedy algorithm. Lastly, we include **BERTopic** (Grootendorst, 2022) which uses class-based TF-IDF on S-BERT document embeddings[4]. For parity reasons, topics comprise the best ten words ($b = 10$) appearing in Wikipedia-V40K.

**Evaluation**. Overall, the topics reclaimed in GPT-2 models are comparable to stronger baselines, suggesting the static interpretations of individual neurons are likely to be meaningful.

[3] http://people.csail.mit.edu/jrennie/20Newsgroups/

[4] all-mpnet-base-v2 S-BERT model.

| modes | Words/Word Pool | | Word Sets/Word Pool | | Words/Word Set | | Empty Pools (%) | |
|---|---|---|---|---|---|---|---|---|
| | *pos* | *neg* | *pos* | *neg* | *pos* | *neg* | *pos* | *neg* |
| 7B-Hi-C | 609.3 ± 323.2 | 39.6 ± 52.9 | 7.22 ± 4.84 | 3.07 ± 3.18 | 13.59 ± 4.29 | 14.10 ± 4.21 | 14.0% | 85.8% |
| 7B-Hi-I | 143.5 ± 64.2 | 143.5 ± 64.4 | 3.60 ± 2.25 | 3.63 ± 2.29 | 13.82 ± 4.26 | 13.68 ± 3.75 | 15.4% | 15.4% |
| 7B-Lo-C | 646.4 ± 314.0 | 33.1 ± 43.6 | 7.51 ± 4.90 | 2.66 ± 2.76 | 13.57 ± 4.26 | 13.81 ± 4.01 | 11.6% | 89.3% |
| 7B-Lo-I | 145.1 ± 85.4 | 142.9 ± 76.5 | 3.80 ± 2.72 | 4.10 ± 2.58 | 13.86 ± 4.29 | 13.77 ± 3.79 | 19.6% | 23.0% |
| 13B-Hi-C | 310.8 ± 121.7 | 100.9 ± 34.7 | 7.35 ± 3.06 | 2.18 ± 1.93 | 14.21 ± 4.63 | 13.04 ± 3.51 | 1.2% | 48.0% |
| 13B-Hi-I | 162.3 ± 38.3 | 162.4 ± 38.6 | 3.97 ± 1.85 | 4.00 ± 1.88 | 13.61 ± 3.98 | 13.51 ± 3.59 | 2.4% | 2.4% |
| 13B-Lo-C | 312.9 ± 123.4 | 99.4 ± 32.2 | 7.39 ± 3.08 | 2.10 ± 1.84 | 14.25 ± 4.69 | 12.96 ± 3.46 | 1.1% | 48.7% |
| 13B-Lo-I | 166.1 ± 50.0 | 158.2 ± 41.5 | 4.13 ± 2.12 | 4.05 ± 2.04 | 13.72 ± 4.13 | 13.52 ± 3.59 | 2.0% | 4.8% |
| Random | 173.7 ± 13.2 | | 4.88 ± 1.30 | | 13.65 ± 3.63 | | 0 | |

Table 5: Sampling statistics on random LLaMA layers. Before the removal of stop topics. Using the Mann-Whitney U test, differences in distributions on the number of Word Sets/Word Pool are significant with $p < 0.001$.

| modes | Mean num. Topics per neuron | | Top-1 Topic (Topical Pool) NPMI score | | Mean NPMI score per topical neurons | | Total num. of Topics | | % of abstract neurons | | |
|---|---|---|---|---|---|---|---|---|---|---|---|
| | *pos* | *neg* | *pos* | *neg* | *pos* | *neg* | *pos* | *neg* | *pos* | *neg* | *both* |
| 7B-Hi-C | 6.98 | 3.07 | 0.128 ± 0.08 | 0.102 ± 0.06 | 0.031 | 0.060 | 2,094,472 | 153,571 | 7.4 | 42.9 | 4.7 |
| 7B-Hi-I | 3.60 | 3.63 | 0.103 ± 0.05 | 0.103 ± 0.05 | 0.050 | 0.050 | 1,071,527 | 1,082,455 | 7.7 | 7.7 | 0.5 |
| 7B-Lo-C | 7.24 | 2.66 | 0.130 ± 0.08 | 0.098 ± 0.06 | 0.030 | 0.060 | 831,424 | 37,233 | 6.2 | 44.7 | 4.1 |
| 7B-Lo-I | 3.80 | 4.10 | 0.101 ± 0.05 | 0.105 ± 0.05 | 0.049 | 0.048 | 400,292 | 413,202 | 9.8 | 11.5 | 0.1 |
| 13B-Hi-C | 7.35 | 2.18 | 0.122 ± 0.05 | 0.054 ± 0.06 | 0.039 | 0.033 | 4,016,410 | 627,028 | 0.6 | 24.0 | 0.0 |
| 13B-Hi-I | 3.97 | 4.00 | 0.094 ± 0.05 | 0.094 ± 0.05 | 0.037 | 0.038 | 2,143,710 | 2,159,556 | 1.2 | 1.2 | 0.1 |
| 13B-Lo-C | 7.39 | 2.10 | 0.122 ± 0.05 | 0.052 ± 0.06 | 0.039 | 0.032 | 1,496,621 | 220,585 | 0.5 | 24.3 | 0.0 |
| 13B-Lo-I | 4.13 | 4.05 | 0.095 ± 0.04 | 0.091 ± 0.04 | 0.038 | 0.037 | 829,311 | 788,671 | 1.0 | 2.4 | 0.0 |
| Random | 4.88 | | 0.099 ± 0.03 | | 0.038 | | 244,145 | | 0 | | |

Table 6: Aggregated results of all LLaMA layers, stop topics excluded. Using the Mann-Whitney U test, differences in the distribution of Top-1 NPMI scores are significant with $p < 0.001$.

The top $K$ extracted topics from GPT-2 models performed the best on 20NewsGroup (Table 4a). At $K = 20$, BBC-News have similar results to 20NewsGroup, and DBLP has similar results to M10 (Table 4b). Evaluating 20NewsGroup and BBC-News at $K = 50$, performance for top $K$ extracted topics is also competitive.

## 8 Extending to LLaMA

**LLaMA** (Touvron et al., 2023) is another popular decoder-only TLM. Its 32K tokens $S_{\text{LLaMA}}$ encoded using SentencePiece's BPE (Kudo and Richardson, 2018). Cross-referencing $S_{\text{LLaMA}}$ with $V_{\text{Wiki}}$ produces few common words, as its tokens are sub-words which are uninterpretable as-is, necessitating additional processing of token pools. Observing the possibility of recovering coherent topics directly from GPT-2's token space and considering a word as a topic of sub-words, we posit that it is possible to recover vocabulary from $S_{\text{LLaMA}}$. We decode each lower-cased $v \in V_{\text{Wiki}}$ via SentencePiece (excluding start token ) into its constituent tokens. Projecting the tokens onto $V_{\text{Wiki}}$, we consider a word recovered when all of its constituent tokens are present in the token pool.

It is possible to recover 36K vocabulary words in $V_{\text{Wiki}}$ from $S_{\text{LLaMA}}$.[5] Using the previous analysis on GPT-2, we examine all layers of LLaMA models sized 7B and 13B.

**Sampling Observations**. Only *pos* pools from complete paths (Hi-C/Lo-C) show a sizeable difference compared to the random baseline. For the various pool from other modes, while the differences are significant, their values are similar or weaker than the random baseline, which suggest they might not be as meaningful.

**Disentangled Topics**. Similar to the observations in sampling, when compared to the random baselines, *pos* pools from complete paths produce more topics and are better in Top-1 $\text{NPMI}_W$. However, pools from the other modes exhibit similar or weaker characteristics, implying that these distributions are indistinguishable. In GPT-2, when compared to their baseline, pools from all modes appear to be meaningful, exhibiting different distributions across metrics. However, results from LLaMA suggest we can only infer and extract topics from *pos* pools produced by complete paths.

---

[5]We exclude multi-word entities representation in $V_{\text{Wiki}}$, as they are joint together using underscores.

| Word pool (alphabetically-sorted) | Word sets | Extracted Topics (NPMI$_W$) | § |
|---|---|---|---|
| acid ais bar baritone bases bass bird blog boer bone broad cap cgi cis closely command commanding composer containing contains contributions dating dean der des die dna dusk editing extra famous fda fisher flag flat floor follows function functionality functionally functioning functions generates geometry germany ghost gis guest gulf hand handing hang host http https integration intentionally jerry lang legs lets like log long longer marker meaning modification mortally naturally noaa non npr ones perry photos plan platform platforms possess post pull realize rna sally sand sas sent slogan soft sport standing stem strategic ted toes touch touching unit upload van weight www (336 ommitted words) | acid bases cap closely containing contains dna editing follows function functionally functioning generates host long marker modification naturally possess stem rna | dna rna bases modification acid stem function naturally containing contains (0.115) | 1 |
| | bird bone broad dusk extra flat floor hand legs like longer meaning ones sand soft standing touch touching weight toes | weight toes hand flat broad legs touch touching bone soft (0.111) | 2 |
| | blog cgi fda http https lang log noaa non npr photos platform sport upload www | cgi www sport photos http https log upload platform blog (0.081) | 3 |
| | baritone command composer contributions des die famous geometry germany guest slogan van der | contributions die command famous germany van des composer guest der (0.048) | 4 |
| | ais bar bass cis functionality functions integration plan platforms unit gis | integration plan bar functionality gis functions ais platforms cis unit (0.001) | 5 |
| | dating dean ghost hang intentionally jerry pull realize sally ted lets | dean pull lets sally hang ghost realize jerry ted dating (-0.053) | 6 |
| | boer fisher flag gulf handing mortally perry post sas sent strategic commanding | post mortally gulf perry flag commanding sas sent handing boer (-0.059) | 7 |

Table 7: LLaMA-13B, Path Hi-C, Layer 3 Neuron 206, *pos* pool. More examples in Appendix C.

**Case Study**. From LLaMA-7B (Table 7), topics consists various themes: cell biology (§1), descriptors (§2), website (§3), mixed (§4), technical (§5). However, some topics seem incoherent, with our best guesses: horror trope (§6) and British military (§7). The size of word sets might hint at some aspect of topic quality.

## 9 Discussions

Aside from comparing across different sizes within the model family, we can also compare across model families. There are differences between the results between GPT-2 and LLaMA. In this section, we discuss the possible cause and effect.

**Topical Sparsity.** The larger sizes of TLMs might induce greater topical sparsity, with larger models able to spread their learnt concepts over a larger number of neurons. With greater topical sparsity, there is greater difficulty in inferring concepts directly from the neuron as its characteristics may be similar to the random baseline. With millions of extracted *pos* pools topics from complete paths of both models, the topics from LLaMA are fewer per neuron and less coherent when compared to GPT-2. Hence, it is likely that the learnt concepts, with similar themes to Wikipedia, are less concentrated within LLaMA.

**Number of Training Tokens**. With LLaMA training on a larger and more diverse corpus, it is likely to learn more concepts absent from Wikipedia. These out-of-domain concepts are undetectable when projected onto $V_{\text{Wiki}}$, causing the neuron to appear less coherent.

**Tokenization.** With LLaMA organizing its knowledge at the token level, reconstructing vocabulary from $S_{\text{LLaMA}}$ introduces another layer of abstraction. This abstraction may result in some informational gap between the word and token level that reduces our ability to infer from the neuron directly.

We believe these are the key challenges to overcome when pursuing prompt-less explanations of TLMs.

## 10 Conclusion

We demonstrate that our approach from a novel angle, disentangling superposed topics from TLM via a graph-based formulation optimizing topic modelling metric NPMI$_W$, revealing and analysing superposed topics on GPT-2, evaluated on ZSTM, and extendable to LLaMA. Additionally, we show that TLM is quantifiable by topic coherence and the number of superposed topics in an automated manner. Comparison using a random baseline gives us confidence when validating interpretations to be meaningful and not due to chance. These metrics might advance our understanding of architectural design decisions and warrant future investigation on other open-sourced TLM.

## Acknowledgments

This research/project is supported by the National Research Foundation, Singapore under its AI Singapore Programme (AISG Award No: AISG3-PhD-2023-08-055T). We thank our reviewers for their kind feedback.

## Limitations

**Language**. Corpora used primarily consist of English vocabulary. The TLMs used are also mostly trained on tokens from English-based corpora. Since our approach is corpus-agnostic, extracting multi-lingual information may be feasible with an appropriate corpus.

**Exact-Solving Time Limit.** Required due to large numbers of sub-graphs. Some sub-graphs fail to solve within the given time limit of 2 or 3 minutes. Nevertheless, we include the sub-optimal topic as the representative topic of the sub-graph.

**Topic Quality.** While $NPMI_W$ is likely to give a good indicator towards the quality of the topic. There is a possibility that some sets of words appear coherent to a group of experts while appearing incomprehensible to others.

## Ethics Statement

We do not foresee any undesired implications stemming from our work. Conversely, we hope that our work can advance AI Ethics research.

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

# A   Supplementary Information

**Algorithm in Words**. We consider each vertex and its neighbours as a *star*. Since $G$ is a complete graph, each star will be similar to other stars. To increase variation among stars, we retain edges with $w_{v_i,v_j} \in [\theta_l, \theta_u]$, deleting all other edges. Next, prioritising stars with larger mean $w_{v_i,v_j}$, greedily select stars with unchosen centres and their remaining unchosen vertices with largest $w_{v_i,v_j}$ fulfilling set size constraints $[\kappa_l, \kappa_u]$. After reducing $G$ to smaller sub-graphs of size 24, it is now computationally feasible to solve these sub-graphs exactly.

## A.1   Hyper-Parameters for GPT-2

For shortlisting of token pools, we set token pool size-parameter $\tau = 900$ for $f^{\mathrm{path}}$ obtained from GPT-2. In our heuristic step, we keep edges with $w_{v_i,v_j}$ that lies between $\theta = [0.1, 1]$. Word-set size constraints to $\kappa = [10, 24]$. GPT-2's tokens can be mapped directly onto Wikipedia's $V$.

## A.2   Zero-Shot Topic Modelling Details

**Corpus size**. Table 8 details the corpora statistics. These corpora may be in the large datasets, such as CommonCrawl, used in the pre-training of PLMs. However, selecting these small corpora would, at best, constitute a tiny percentage of the total tokens used to train these PLMs. Furthermore, some of our neural topic model baselines will also utilise PLM embeddings. When possible, we try to mitigate the influence of rare word pairs, based on token count and corpus size, by setting the minimum count of word pairs for $\mathrm{NPMI}_C$ calculation for M10, DBLP, 20NewsGroup and BBC-News to 0, 5, 50, 50 respectively.

| Corpus | |Docs| | |Tokens| | $|V|$ | $|V| \in$ Wiki (%) |
|---|---|---|---|---|
| 20NewsGroup | 16,309 | 783,151 | 1,612 | 1,582 (98.1) |
| BBC-News | 2,225 | 267,259 | 2,949 | 2,892 (98.1) |
| M10 | 8,355 | 49,385 | 1,696 | 1,613 (95.1) |
| DBLP | 54,595 | 294,757 | 1,513 | 1,394 (92.1) |

Table 8: Numerical descriptions of corpora.

**Hyper-Parameters**. For GPT-2, we set $\tau = 900$. Table 9 contains the $\theta$ hyper-parameter for edge-weights. $\theta_l$ have to be reduced below 0 to produce stars sub-graphs. For M10 and DBLP, their small document size and rare word pairs strongly influence their $\mathrm{NPMI}_C$. To mitigate this, we reduce $\theta_u$. We set a minimum word-pair count that will retain a majority of words such that they contain at least a word pair.

| Corpus | $\theta$ edge-weight | Sub-graph size |
|---|---|---|
| 20NewsGroup | [-0.6, 1.0] | 16 |
| BBC-News | [-0.6, 1.0] | 16 |
| DBLP | [-0.1, 0.3] | 32 |
| M10 | [-0.1, 0.3] | 16 |

Table 9: Hyper-parameters used in heuristic pruning of NPMI graph for ZSTM.

**Voting Scheme**. We give each token ten votes per layer, awarded to the highest activated neuron. Neurons that exceed the vote count threshold will be examined, with the neuron acting as the origin. The vote count threshold is set relative to corpus size, at 25% for 20NewsGroup and BBC-News, and 1% for M10 and DBLP.

**Greedy Algorithm**. From the extracted topics, we choose $K$ best topics greedily using heuristics prioritising for $\mathrm{NPMI}_C$ and modulated by $\epsilon$, which represents the maximum similar words between each topic. We search $1 \leq \epsilon \leq 3$ and select the most competitive.

| Corpus | K | ProdLDA-$\beta$ | CTM-$\beta$ | GPT-2 |
|---|---|---|---|---|
| 20NewsGroup | 20 | 2 | 1 | 2 |
| BBC-News | 20 | 2 | 3 | 2 |
| DBLP | 20 | 2 | 3 | 2 |
| M10 | 20 | 3 | 3 | 2 |
| 20NewsGroup | 50 | 3 | 3 | 2 |
| BBC-News | 50 | 3 | 3 | 2 |

Table 10: Hyper-parameter $\epsilon$ for greedy-algo.

## A.3   Additional LLaMA Details

**Hyper-parameters**. To shortlist token pools, we set size-parameter $\tau = 900$ for $f^{\mathrm{path}}$ obtained from LLaMA. In our heuristic step, we keep edges with $w_{v_i,v_j}$ that lies between $\theta = [0.025, 1]$. Word-set size constrained to $\kappa_{\mathrm{pos}} = [10, 32]$, $\kappa_{\mathrm{neg}} = [10, 24]$. A larger upper-limit reduces number of word-sets generated.

**Word Explosion**. LLaMA's pool can reconstruct a larger number of words from a smaller set of tokens. To prevent this scenario, we automatically reduce $\tau$, 50 at a time, until the word pools generated have less than 900 words.

# B Supplementary Tables

| modes | Words/Word Pool | | Word Sets/Word Pool | | Words/Word Set | |
|---|---|---|---|---|---|---|
| | *pos* | *neg* | *pos* | *neg* | *pos* | *neg* |
| S-Hi-C | 388.1 ± 122.1 | 216.680 ± 87.9 | 6.90 ± 5.43 | 2.07 ± 1.89 | 15.19 ± 4.84 | 12.82 ± 3.30 |
| S-Hi-I | 480.0 ± 104.9 | 229.119 ± 87.0 | 9.86 ± 5.79 | 2.19 ± 2.22 | 14.37 ± 4.26 | 12.65 ± 3.21 |
| S-Lo-C | 326.4 ± 125.2 | 228.968 ± 86.1 | 5.92 ± 5.00 | 1.70 ± 1.11 | 15.10 ± 4.76 | 12.49 ± 2.91 |
| S-Lo-I | 437.3 ± 113.1 | 336.300 ± 114.1 | 8.11 ± 5.18 | 3.56 ± 2.79 | 14.47 ± 4.38 | 12.74 ± 3.14 |
| M-Hi-C | 357.2 ± 98.4 | 262.420 ± 94.0 | 5.45 ± 3.85 | 2.69 ± 3.01 | 14.72 ± 4.63 | 12.74 ± 3.22 |
| M-Hi-I | 510.5 ± 103.2 | 338.652 ± 93.3 | 11.03 ± 5.29 | 3.87 ± 2.87 | 14.41 ± 4.25 | 12.93 ± 3.30 |
| M-Lo-C | 346.0 ± 113.5 | 268.337 ± 101.8 | 5.61 ± 4.44 | 2.11 ± 2.01 | 14.65 ± 4.55 | 12.27 ± 2.73 |
| M-Lo-I | 454.9 ± 95.9 | 383.143 ± 99.5 | 6.79 ± 4.43 | 4.87 ± 3.41 | 13.81 ± 3.98 | 12.96 ± 3.29 |
| L-Hi-C | 409.9 ± 114.6 | 267.437 ± 82.1 | 8.42 ± 4.70 | 3.08 ± 4.30 | 14.43 ± 4.32 | 13.76 ± 4.09 |
| L-Hi-I | 473.7 ± 88.8 | 211.120 ± 110.7 | 9.27 ± 3.89 | 4.77 ± 5.31 | 14.40 ± 4.50 | 14.10 ± 4.19 |
| L-Lo-C | 408.4 ± 94.7 | 248.464 ± 75.9 | 7.79 ± 4.68 | 2.41 ± 3.31 | 14.43 ± 4.35 | 13.14 ± 3.70 |
| L-Lo-I | 449.7 ± 92.2 | 340.385 ± 96.9 | 6.66 ± 3.97 | 3.96 ± 3.18 | 13.74 ± 3.94 | 12.89 ± 3.27 |
| XL-Hi-C | 473.0 ± 129.8 | 250.104 ± 83.9 | 11.34 ± 5.29 | 3.08 ± 4.09 | 14.88 ± 4.55 | 13.66 ± 4.00 |
| XL-Hi-I | 478.2 ± 96.0 | 244.775 ± 106.4 | 10.02 ± 3.73 | 3.84 ± 5.10 | 14.60 ± 4.58 | 14.00 ± 4.17 |
| XL-Lo-C | 466.4 ± 102.6 | 244.433 ± 79.7 | 10.02 ± 5.06 | 2.48 ± 3.64 | 14.78 ± 4.51 | 13.44 ± 3.96 |
| XL-Lo-I | 450.4 ± 89.2 | 351.506 ± 106.3 | 6.88 ± 4.10 | 3.99 ± 3.55 | 13.76 ± 3.94 | 13.01 ± 3.42 |
| Random | 559.2 ± 14.2 | | 5.77 ± 1.87 | | 12.04 ± 2.37 | |

Table 11: Sampling statistics from various paths of GPT-2. All neurons of respective modes produced non-empty word pools. We conduct the Mann-Whitney U test between the random baseline and non-random methods on the number of word sets/neuron, and their differences are significant with $p < 0.001$.

| modes | Mean num. Topics | | Top-1 Topic (Topical Pool) NPMI$_W$ score | | Mean NPMI$_W$ score per Topical Pool | | Total num. of topics | | % of Abstract Pool | | |
|---|---|---|---|---|---|---|---|---|---|---|---|
| | *pos* | *neg* | *pos* | *neg* | *pos* | *neg* | *pos* | *neg* | *pos* | *neg* | *both* |
| S-Hi-C | 6.76 | 2.08 | 0.202 ± 0.06 | 0.216 ± 0.09 | 0.130 | 0.164 | 211,612 | 22,106 | 7.5 | 35.6 | 6.0 |
| S-Hi-I | 9.76 | 2.19 | 0.218 ± 0.06 | 0.166 ± 0.08 | 0.119 | 0.132 | 351,968 | 24,634 | 1.1 | 34.8 | 0.6 |
| S-Lo-C | 5.68 | 1.70 | 0.215 ± 0.07 | 0.199 ± 0.09 | 0.142 | 0.162 | 40,865 | 4,725 | 11.0 | 34.9 | 7.4 |
| S-Lo-I | 8.10 | 3.56 | 0.227 ± 0.07 | 0.191 ± 0.08 | 0.122 | 0.117 | 70,502 | 21,913 | 2.8 | 16.6 | 1.2 |
| M-Hi-C | 4.89 | 2.70 | 0.192 ± 0.06 | 0.184 ± 0.08 | 0.128 | 0.133 | 405,662 | 88,914 | 7.8 | 33.2 | 4.2 |
| M-Hi-I | 10.84 | 3.80 | 0.232 ± 0.06 | 0.162 ± 0.06 | 0.119 | 0.108 | 1,048,594 | 270,000 | 0.8 | 13.8 | 0.4 |
| M-Lo-C | 5.45 | 2.12 | 0.209 ± 0.07 | 0.169 ± 0.08 | 0.134 | 0.122 | 108,242 | 16,967 | 9.6 | 33.7 | 6.0 |
| M-Lo-I | 6.80 | 4.87 | 0.216 ± 0.07 | 0.190 ± 0.07 | 0.114 | 0.111 | 158,952 | 97,378 | 2.4 | 9.3 | 0.3 |
| L-Hi-C | 8.26 | 3.04 | 0.221 ± 0.06 | 0.165 ± 0.08 | 0.130 | 0.126 | 1405,522 | 214,319 | 3.9 | 30.9 | 1.3 |
| L-Hi-I | 8.37 | 4.69 | 0.188 ± 0.05 | 0.180 ± 0.06 | 0.111 | 0.128 | 1,469,191 | 241,072 | 2.4 | 36.1 | 0.2 |
| L-Lo-C | 7.76 | 2.41 | 0.226 ± 0.06 | 0.164 ± 0.08 | 0.136 | 0.126 | 338,917 | 39,909 | 2.6 | 32.1 | 1.7 |
| L-Lo-I | 6.66 | 3.96 | 0.208 ± 0.07 | 0.186 ± 0.07 | 0.111 | 0.114 | 291,240 | 133,769 | 2.6 | 13.4 | 0.4 |
| XL-Hi-C | 10.88 | 2.98 | 0.232 ± 0.07 | 0.160 ± 0.07 | 0.127 | 0.126 | 3,135,250 | 291,738 | 3.1 | 34.0 | 0.6 |
| XL-Hi-I | 9.12 | 3.78 | 0.199 ± 0.05 | 0.160 ± 0.06 | 0.117 | 0.118 | 2,665,849 | 535,733 | 2.4 | 26.9 | 0.0 |
| XL-Lo-C | 9.62 | 2.47 | 0.227 ± 0.07 | 0.155 ± 0.07 | 0.128 | 0.126 | 712,337 | 59,817 | 1.8 | 34.2 | 0.9 |
| XL-Lo-I | 6.88 | 4.00 | 0.212 ± 0.06 | 0.182 ± 0.07 | 0.111 | 0.113 | 508,179 | 217,321 | 1.9 | 14.6 | 0.3 |
| Random | 5.77 | | 0.097 ± 0.03 | | 0.029 | | 261,949 | | 0 | | |

Table 12: Exploratory results from mining topics in GPT-2 with "stop topics" removed. We conduct the Mann-Whitney U test between random baseline and non-random methods on top-1 NPMI$_W$ scores. Differences between GPT-2-sampled and randomly-sampled for both metrics are significant with $p < 0.001$. Abstract neurons are neurons of respective paths that have no projected topics.

| Model | NPMI$_C$ | NPMI$_W$ | TU | $A$ (%) |
|---|---|---|---|---|
| ProdLDA | -0.135 | 0.021 | 0.750 | - |
| ProdLDA-$\beta$ | 0.018 | 0.061 | 0.796 | - |
| CTM | -0.100 | 0.044 | _0.825_ | - |
| CTM-$\beta$ | _0.048_ | _0.075_ | 0.809 | - |
| BERTopic | -0.045 | 0.064 | 0.799 | - |
| GPT-2-S-Lo-C | **0.134** | **0.074** | **0.870** | 20.57 |
| GPT-2-S-Lo-I | **0.151** | 0.066 | **0.866** | 20.57 |
| GPT-2-M-Lo-C | **0.156** | **0.075** | **0.864** | 23.73 |
| GPT-2-M-Lo-I | **0.155** | **0.082** | **0.858** | 23.73 |
| GPT-2-L-Lo-C | **0.164** | **0.077** | **0.862** | 14.99 |
| GPT-2-L-Lo-I | **0.155** | **0.079** | **0.864** | 14.99 |
| GPT-2-XL-Lo-C | **0.171** | **0.076** | **0.860** | 15.37 |
| GPT-2-XL-Lo-I | **0.171** | **0.090** | **0.866** | 15.37 |

(a) 20NewsGroup, $K = 50$

| Model | NPMI$_C$ | NPMI$_W$ | TU | $A$ (%) |
|---|---|---|---|---|
| ProdLDA | -0.131 | 0.026 | 0.667 | - |
| ProdLDA-$\beta$ | -0.108 | 0.044 | _0.768_ | - |
| CTM | -0.118 | 0.048 | 0.701 | - |
| CTM-$\beta$ | _-0.088_ | _0.056_ | 0.767 | - |
| BERTopic | - | - | - | - |
| GPT-2-S-Lo-C | **0.100** | **0.056** | **0.852** | 39.56 |
| GPT-2-S-Lo-I | **0.111** | **0.063** | **0.838** | 39.56 |
| GPT-2-M-Lo-C | **0.111** | **0.051** | **0.856** | 44.71 |
| GPT-2-M-Lo-I | **0.113** | **0.057** | **0.870** | 44.71 |
| GPT-2-L-Lo-C | **0.118** | **0.062** | **0.840** | 28.96 |
| GPT-2-L-Lo-I | **0.115** | **0.056** | **0.848** | 28.96 |
| GPT-2-XL-Lo-C | **0.123** | **0.058** | **0.858** | 31.74 |
| GPT-2-XL-Lo-I | **0.122** | **0.071** | **0.836** | 31.74 |

(b) BBC-News, $K = 50$. BERTopic unable to produce 50 topics for all runs.

Table 13: Results for topic set sizes $K = 50$. The best baseline scores for the metric are underlined. Bolded PLM scores are better than the best baseline. Baseline scores are the mean of 10 independent runs with error bars in Table 15 in Appendix B. The minimum word co-occurrence count is 50 for NPMI$_C$ calculation.

| Model | NPMI$_C$ | NPMI$_W$ | TU | $A$ (%) |
|---|---|---|---|---|
| ProdLDA | -0.059 | 0.045 | 0.903 | - |
| ProdLDA-$\beta$ | _0.085_ | _0.075_ | 0.911 | - |
| CTM | -0.060 | 0.052 | 0.916 | - |
| CTM-$\beta$ | 0.016 | 0.070 | _0.945_ | - |
| BERTopic | -0.002 | 0.069 | 0.887 | - |
| GPT-2-S-Lo-C | **0.164** | **0.123** | **0.955** | 20.57 |
| GPT-2-S-Lo-I | **0.169** | **0.115** | **0.950** | 20.57 |
| GPT-2-M-Lo-C | **0.177** | **0.117** | **0.965** | 23.73 |
| GPT-2-M-Lo-I | **0.186** | **0.120** | 0.920 | 23.73 |
| GPT-2-L-Lo-C | **0.197** | **0.115** | **0.945** | 14.99 |
| GPT-2-L-Lo-I | **0.170** | **0.118** | **0.965** | 14.99 |
| GPT-2-XL-Lo-C | **0.184** | **0.112** | **0.940** | 15.37 |
| GPT-2-XL-Lo-I | **0.179** | **0.140** | **0.955** | 15.37 |

(a) 20NewsGroup, $K = 20$. Minimum word co-occurence count is 50 for NPMI$_C$ calculation.

| Model | NPMI$_C$ | NPMI$_W$ | TU | $A$ (%) |
|---|---|---|---|---|
| ProdLDA | -0.085 | 0.027 | 0.850 | - |
| ProdLDA-$\beta$ | -0.069 | 0.059 | _0.888_ | - |
| CTM | -0.052 | 0.053 | 0.828 | - |
| CTM-$\beta$ | _0.009_ | 0.072 | 0.818 | - |
| BERTopic | -0.036 | _0.077_ | 0.862 | - |
| GPT-2-S-Lo-C | **0.139** | **0.119** | **0.920** | 39.56 |
| GPT-2-S-Lo-I | **0.135** | **0.108** | **0.900** | 39.56 |
| GPT-2-M-Lo-C | **0.147** | **0.106** | **0.915** | 44.71 |
| GPT-2-M-Lo-I | **0.141** | **0.123** | **0.920** | 44.71 |
| GPT-2-L-Lo-C | **0.139** | **0.116** | **0.920** | 28.96 |
| GPT-2-L-Lo-I | **0.147** | **0.114** | **0.920** | 28.96 |
| GPT-2-XL-Lo-C | **0.148** | **0.112** | **0.900** | 31.74 |
| GPT-2-XL-Lo-I | **0.143** | **0.122** | **0.920** | 31.74 |

(b) BBC-News, $K = 20$. Minimum word co-occurence count is 50 for NPMI$_C$ calculation.

| Model | NPMI$_C$ | NPMI$_W$ | TU | $A$ (%) |
|---|---|---|---|---|
| ProdLDA | -0.041 | 0.024 | _0.897_ | - |
| ProdLDA-$\beta$ | -0.061 | 0.025 | 0.886 | - |
| CTM | -0.035 | 0.034 | 0.863 | - |
| CTM-$\beta$ | _-0.001_ | _0.050_ | 0.812 | - |
| BERTopic | -0.212 | -0.024 | 0.791 | - |
| GPT-2-S-Lo-C | **0.082** | 0.028 | 0.7 . | 28.63 |
| GPT-2-S-Lo-I | **0.075** | 0.038 | 0.780 | 28.63 |
| GPT-2-M-Lo-C | **0.087** | 0.035 | 0.800 | 43.12 |
| GPT-2-M-Lo-I | **0.081** | 0.040 | 0.800 | 43.12 |
| GPT-2-L-Lo-C | **0.091** | 0.042 | 0.800 | 34.18 |
| GPT-2-L-Lo-I | **0.085** | 0.039 | 0.845 | 34.18 |
| GPT-2-XL-Lo-C | **0.103** | 0.033 | 0.850 | 38.98 |
| GPT-2-XL-Lo-I | **0.088** | **0.057** | 0.845 | 38.98 |

(c) DBLP, $K = 20$. Minimum word co-occurence count is 5 for NPMI$_C$ calculation.

| Model | NPMI$_C$ | NPMI$_W$ | TU | $A$ (%) |
|---|---|---|---|---|
| ProdLDA | -0.028 | 0.007 | 0.741 | - |
| ProdLDA-$\beta$ | -0.023 | 0.014 | 0.830 | - |
| CTM | 0.027 | 0.050 | 0.895 | - |
| CTM-$\beta$ | _0.067_ | _0.057_ | 0.835 | - |
| BERTopic | -0.049 | 0.023 | _0.906_ | - |
| GPT-2-S-Lo-C | **0.135** | 0.021 | 0.850 | 32.31 |
| GPT-2-S-Lo-I | **0.143** | 0.043 | 0.885 | 32.31 |
| GPT-2-M-Lo-C | **0.156** | 0.021 | 0.865 | 46.52 |
| GPT-2-M-Lo-I | **0.148** | 0.042 | 0.865 | 46.52 |
| GPT-2-L-Lo-C | **0.164** | 0.028 | 0.855 | 38.20 |
| GPT-2-L-Lo-I | **0.143** | 0.033 | 0.855 | 38.20 |
| GPT-2-XL-Lo-C | **0.180** | 0.023 | 0.845 | 43.39 |
| GPT-2-XL-Lo-I | **0.159** | 0.045 | 0.850 | 43.39 |

(d) M10, $K = 20$. Minimum word co-occurence count is 0 for NPMI$_C$ calculation due to low token and document count.

Table 14: Results for topic set sizes $K = 20$. The best baseline scores for the metric are underlined. Bolded PLM scores are better than the best baseline. The mean of 10 independent runs with error bars in Table 15 in Appendix B.

|  | NPMI$_C$ | NPMI$_W$ | TU |
|---|---|---|---|
| ProdLDA | -0.059 ± 0.014 | 0.045 ± 0.007 | 0.903 ± 0.021 |
| ProdLDA-$\beta$ | 0.085 ± 0.021 | 0.075 ± 0.007 | 0.911 ± 0.010 |
| CTM | -0.060 ± 0.024 | 0.052 ± 0.008 | 0.916 ± 0.012 |
| CTM-$\beta$ | 0.016 ± 0.019 | 0.070 ± 0.007 | 0.945 ± 0.007 |
| BERTopic | -0.002 ± 0.023 | 0.069 ± 0.007 | 0.887 ± 0.012 |

(a) 20NewsGroup, $K = 20$

|  | NPMI$_C$ | NPMI$_W$ | TU |
|---|---|---|---|
| ProdLDA | -0.085 ± 0.025 | 0.027 ± 0.010 | 0.850 ± 0.035 |
| ProdLDA-$\beta$ | -0.069 ± 0.018 | 0.059 ± 0.011 | 0.888 ± 0.016 |
| CTM | -0.052 ± 0.014 | 0.053 ± 0.013 | 0.828 ± 0.041 |
| CTM-$\beta$ | 0.009 ± 0.019 | 0.072 ± 0.006 | 0.818 ± 0.015 |
| BERTopic | -0.036 ± 0.012 | 0.077 ± 0.007 | 0.862 ± 0.008 |

(b) BBC-News, $K = 20$

|  | NPMI$_C$ | NPMI$_W$ | TU |
|---|---|---|---|
| ProdLDA | -0.041 ± 0.015 | 0.024 ± 0.004 | 0.897 ± 0.013 |
| ProdLDA-$\beta$ | -0.061 ± 0.018 | 0.025 ± 0.008 | 0.886 ± 0.009 |
| CTM | -0.035 ± 0.013 | 0.034 ± 0.009 | 0.863 ± 0.019 |
| CTM-$\beta$ | -0.001 ± 0.011 | 0.050 ± 0.007 | 0.812 ± 0.020 |
| BERTopic | -0.212 ± 0.016 | -0.024 ± 0.004 | 0.791 ± 0.018 |

(c) DBLP, $K = 20$

|  | NPMI$_C$ | NPMI$_W$ | TU |
|---|---|---|---|
| ProdLDA | -0.028 ± 0.024 | 0.007 ± 0.010 | 0.741 ± 0.027 |
| ProdLDA-$\beta$ | -0.023 ± 0.020 | 0.014 ± 0.011 | 0.830 ± 0.021 |
| CTM | 0.027 ± 0.013 | 0.050 ± 0.009 | 0.895 ± 0.026 |
| CTM-$\beta$ | 0.067 ± 0.028 | 0.057 ± 0.009 | 0.835 ± 0.015 |
| BERTopic | -0.049 ± 0.018 | 0.023 ± 0.012 | 0.906 ± 0.011 |

(d) M10, $K = 20$

|  | NPMI$_C$ | NPMI$_W$ | TU |
|---|---|---|---|
| ProdLDA | -0.135 ± 0.012 | 0.021 ± 0.005 | 0.750 ± 0.024 |
| ProdLDA-$\beta$ | 0.018 ± 0.012 | 0.061 ± 0.003 | 0.796 ± 0.010 |
| CTM | -0.100 ± 0.014 | 0.044 ± 0.006 | 0.825 ± 0.022 |
| CTM-$\beta$ | 0.048 ± 0.006 | 0.075 ± 0.004 | 0.809 ± 0.007 |
| BERTopic | -0.045 ± 0.008 | 0.064 ± 0.002 | 0.799 ± 0.012 |

(e) 20NewsGroup, $K = 50$.

|  | NPMI$_C$ | NPMI$_W$ | TU |
|---|---|---|---|
| ProdLDA | -0.131 ± 0.016 | 0.026 ± 0.006 | 0.667 ± 0.033 |
| ProdLDA-$\beta$ | -0.108 ± 0.016 | 0.044 ± 0.011 | 0.768 ± 0.009 |
| CTM | -0.118 ± 0.025 | 0.048 ± 0.006 | 0.701 ± 0.051 |
| CTM-$\beta$ | -0.088 ± 0.010 | 0.056 ± 0.005 | 0.767 ± 0.008 |

(f) BBC-News, $K = 50$. BERTopic could not produce enough topics for all 10 independent run and is excluded.

Table 15: Baseline results with error bars. Underline scores are best within metric.

| | Neurons/layer | Layers | Total Neurons |
|---|---|---|---|
| **GPT-2** | | | |
| S-Hi-C | 3,072 | 12 | 36,864 |
| S-Hi-I | 3,072 | 12 | 36,864 |
| S-Lo-C | 768 | 12 | 9,216 |
| S-Lo-I | 768 | 12 | 9,216 |
| M-Hi-C | 4,096 | 24 | 98,304 |
| M-Hi-I | 4,096 | 24 | 98,304 |
| M-Lo-C | 1,024 | 24 | 24,576 |
| M-Lo-I | 1,024 | 24 | 24,576 |
| L-Hi-C | 5,120 | 36 | 184,320 |
| L-Hi-I | 5,120 | 36 | 184,320 |
| L-Lo-C | 1,280 | 36 | 46,080 |
| L-Lo-I | 1,280 | 36 | 46,080 |
| XL-Hi-C | 6,400 | 48 | 307,200 |
| XL-Hi-I | 6,400 | 48 | 307,200 |
| XL-Lo-C | 1,600 | 48 | 76,800 |
| XL-Lo-I | 1,600 | 48 | 76,800 |
| **LLaMA** | | | |
| 7B-Hi-C | 11,008 | 32 | 352,256 |
| 7B-Hi-I | 11,008 | 32 | 352,256 |
| 7B-Lo-C | 4,096 | 32 | 131,072 |
| 7B-Lo-I | 4,096 | 32 | 131,072 |
| 13B-Hi-C | 13,824 | 40 | 552,960 |
| 13B-Hi-I | 13,824 | 40 | 552,960 |
| 13B-Lo-C | 5,120 | 40 | 204,800 |
| 13B-Lo-I | 5,120 | 40 | 204,800 |

Table 16: Model parameters statistics. For Paths Hi-C and Hi-I, the neurons refer to those in the transformer block's MLP, usually containing more neurons than Paths Lo-C and Lo-I, located in the residual stream.

## C Examples

Words pools are alphabetically-sorted.

| Word pool | Word sets | Extracted Topics (NPMI$_W$) | # |
|---|---|---|---|
| archbishop bishop canterbury cathedral conversion copenhagen danish doctrine earthquake helsinki leicester lutheran mercy norse norwegian norwich orthodox oslo pilgrim pope premiership repentance resurrection sacrament salvation scandinavian sins souls stockholm sweden swedish viking worcester (332 ommitted words) | lutheran oslo danish stockholm viking norse norwegian sweden swedish helsinki copenhagen scandinavian | copenhagen swedish oslo norwegian helsinki stockholm viking danish scandinavian sweden (0.237) | 1 |
| | salvation souls doctrine orthodox resurrection mercy sins conversion sacrament repentance | salvation souls doctrine repentance sins mercy conversion sacrament resurrection orthodox (0.179) | 2 |
| | archbishop pope worcester premiership earthquake leicester norwich cathedral pilgrim bishop canterbury | archbishop bishop pope earthquake pilgrim canterbury norwich worcester cathedral leicester (0.098) | 3 |

Table 17: GPT-2-XL, Path Hi-C, Layer 26 Neuron 5201, *neg* pool. Vikings (#1) and Religion (#2, 3).

| Word pool | Word sets | Extracted Topics (NPMI$_W$) | # |
|---|---|---|---|
| answer array bat beat bet better boot certainly client come command control correctly crowd daily day decided don drive exactly february feel feeling flip folks friday fun function functions getting going good got guys install installed know let like little module modules monday month morning need night ought pass people pretty saturday smart sort sunday tell throw throws thursday tie tonight toss tossed tracking tuesday understand wednesday (358 ommitted words) | wednesday saturday tuesday monday month morning february good daily day night thursday sunday tonight friday | sunday morning monday wednesday night friday saturday thursday day tuesday (0.331) | 1 |
| | don let guys exactly going tell feel ought understand need answer getting folks better come pretty like certainly people sort feeling fun little know | let pretty tell going like fun don feel know guys (0.179) | 2 |
| | modules boot array client command install installed tracking control smart function functions module | module modules client smart control functions install tracking installed array (0.114) | 3 |
| | correctly crowd drive got decided pass flip bat throw throws tie beat tossed bet toss | bat throws drive tossed got crowd pass tie toss throw (0.109) | 4 |

Table 18: GPT-2-XL, Path Hi-C, Layer 26 Neuron 5201, *pos* pool. Stop topic (#1), relationship-related (#2), software (#3), and baseball/cricket (#4).

| Word pool | Word sets | Extracted Topics (NPMI$_W$) | # |
|---|---|---|---|
| aluminum chemist foods magnesium mineral oxide phosphate phosphorus salts soda thermal uranium vapor (251 ommitted words) | uranium magnesium chemist oxide thermal phosphate phosphorus vapor foods mineral aluminum soda salts | aluminum mineral magnesium soda phosphate phosphorus uranium salts oxide thermal (0.183) | 1 |

Table 19: GPT-2-XL, Path Lo-C, Layer 2 Neuron 613, *neg* pool. Some pools only have one topic. Chemistry (#1).

| Word pool | Word sets | Extracted Topics (NPMI$_W$) | # |
|---|---|---|---|
| access adam advertising alex andy answer app apr ass author ben blog book chapter check com content contents dear dec don download editor email facebook features file google got help hey hit http https install jan jason jon josh kill let like love mar media michael net news notes nov online page phil photo picture post posted preview print published quote read reddit rick said sam sep server source story subscribe summary ted text thank tom trump twitter update updated user version video watch web wow written www yes (213 ommitted words) | dec com web nov jan www sep mar https http apr | www apr jan dec sep https nov http mar com (0.317) | 1 |
| | page twitter google posted reddit app media post email blog user photo online update content video news advertising watch download preview updated trump facebook | twitter page content reddit online facebook posted user app google (0.246) | 2 |
| | tom ben phil andy adam michael alex ted jason rick help kill jon josh sam | phil michael sam andy kill alex josh ben jason tom (0.199) | 3 |
| | chapter read story contents quote notes written author picture published text summary editor book | contents chapter read written published author story summary text book (0.196) | 4 |
| | hit let wow got answer love like yes ass said hey dear thank don | like let dear got thank answer said don love yes (0.131) | 5 |
| | access install check source net features subscribe print version file server | net access source version check install file features server print (0.070) | 6 |

Table 20: GPT-2-XL, Path Lo-C, Layer 2 Neuron 613, *pos* pool. Stop topic (#1), due to "months", social media (#2), conversational (#3), names (#4), story (#5), and software (#6).

| Word pool | Word sets | Extracted Topics (NPMI$_W$) | # |
|---|---|---|---|
| absolutely access allowing amazing ancient app available awesome basic beat beautiful best better bit blood casting certainly character common convenient crazy critical definitely difficult digital direct don dream easily easy effective experience extreme extremely feel feeling final free friends fun going good got happy heart highly hit hopefully hot incredible information intriguing jump kind know like little look lost lot maybe memory negative number pain perfect performance pop positive precious pretty quick rare reach resource resources rich role sad safe said second single sort spending stuff stunning success super sure sweet team thing things think thinking thought times totally touch treasure truly useful usually valuable version virtual won wonderful writing yes (423 ommitted words) | don maybe know lot definitely said pretty bit feel going hopefully kind absolutely sort sure wonderful fun good awesome certainly yes got like think | lot bit definitely think hopefully maybe fun don feel know (0.218) | 1 |
| | hit pop crazy spending extremely easy stuff version single number digital sweet reach hot | easy pop sweet stuff hot single version crazy hit number (0.122) | 2 |
| | character truly look dream incredible stunning thought perfect happy casting totally friends sad amazing | perfect truly totally sad character amazing happy thought look incredible (0.105) | 3 |
| | team lost jump second easily final beat times super best won | beat jump won super best lost times team second final (0.096) | 4 |
| | available basic allowing information memory virtual quick better difficult free convenient direct safe app access | app access basic available allowing free direct information memory virtual (0.093) | 5 |
| | little blood resource resources valuable thing things rich beautiful ancient rare treasure precious | ancient valuable treasure little rare precious rich beautiful thing things (0.090) | 6 |
| | highly positive writing negative thinking success performance role intriguing critical | writing performance positive negative intriguing critical role highly success thinking (0.081) | 7 |
| | experience extreme feeling effective useful usually common touch heart pain | heart experience useful effective common touch extreme pain feeling usually (0.053) | 8 |

Table 21: GPT-2-XL, Path Hi-I, Layer 26 Neuron 5201, *pos* pool. Conversational (#1), pop music (#2), story-related (#3), sports (#4), software-related (#5), exploration (#6), art-related (#7), and medical-related (#8).

| Word pool | Word sets | Extracted Topics (NPMI$_W$) | # |
|---|---|---|---|
| air alert black broad carbon cycle dam dark deck deep electric electrical energy gas global heat height high higher horizontal iron light long low narrow negative oxygen painted power range red reduced steel steep structural supply tape thick tight water white wide zero (322 ommitted words) | gas low dam electric electrical energy supply heat water power | supply electric electrical energy heat dam water gas low power (0.161) | 1 |
| | high long broad steep narrow deck thick tight height range wide | narrow range high thick deck steep long wide broad height (0.146) | 2 |
| | higher reduced global oxygen cycle steel zero air negative structural iron carbon | oxygen higher structural iron carbon steel negative cycle reduced zero (0.091) | 3 |
| | tape white horizontal painted dark deep alert light black red | black white red light tape alert horizontal deep dark painted (0.080) | 4 |

Table 22: GPT-2-XL, Path Lo-I, Layer 38 Neuron 556, *pos* pool. Power generation (#1), Visuals (#2, 4), and chemistry (#3).

| Word pool | Word sets | Extracted Topics (NPMI$_W$) | # |
|---|---|---|---|
| angel birds breeding canary cats champagne coastline crop dams deities demon demons descend devils dogs dragons drought eggs farmers flora glaciers grapes growers guide horses hunt lake lakes land landscapes lion lions maize mill peaks plateau pond ponds reptiles river rivers roses south southern spirits strawberries tails tasmania trout valleys vampires waters watershed wind wolves (310 ommitted words) | waters watershed tasmania trout pond ponds drought lake valleys lakes river rivers mill dams | lakes river dams pond ponds rivers waters watershed lake trout (0.321) | 1 |
| | south southern wind coastline land flora landscapes peaks plateau glaciers | southern flora coastline wind landscapes plateau land peaks south glaciers (0.109) | 2 |
| | crop breeding farmers maize grapes eggs growers roses champagne strawberries | breeding eggs strawberries champagne roses maize grapes crop farmers growers (0.065) | 3 |
| | spirits dragons hunt vampires deities demon devils angel cats demons | devils cats hunt spirits dragons vampires deities angel demon demons (0.044) | 4 |
| | tails horses wolves reptiles guide canary descend birds lion lions dogs | tails guide lion lions horses reptiles birds dogs descend wolves (0.021) | 5 |

Table 23: GPT-2-XL, Path Lo-I, Layer 13 Neuron 1126, *neg* pool. Nature (#1, 2), agriculture (#3), magical beasts (#4), and animals (#5).

| Word pool | Word sets | Extracted Topics (NPMI$_W$) | # |
|---|---|---|---|
| acceleration actors alive allow allows angry arrow atoms break breath broke calculations camera case com combine computation connection connections continued convenient cop crash crashed crowd deliver denotes dependence depth detail details distance editing entropy estimation explain explained explanation facing fall fast figure flash formula function functions glass ground half heavy hooked http https hundreds icons identify including inserting large lead leads leave libraries like likewise manipulate masses mechanical media met miles minus net object objected objective official pad pages particular passes passing pause periods play played possible pressed privileges programs promises provides recursive saving scripted showed shown shut skip slight starting stating step talk temporary terrain thirty today typical understanding upload user users victim videos violent wanting watch words www yards zero (111 ommitted words) | com connection connections detail details http libraries media net official pages today upload user users videos watch www https | pages com www user users http watch https upload videos (0.230) | 1 |
| | allow allows case computation depth estimation formula function functions leads like object particular possible programs provides showed shown starting step words zero recursive | recursive formula estimation computation object allows function step functions zero (0.146) | 2 |
| | broke crashed crowd distance facing ground half hooked hundreds including lead miles minus passes passing play played shut thirty yards | distance ground lead crowd broke miles half passes passing yards (0.087) | 3 |
| | arrow atoms calculations denotes dependence explain explained explanation glass identify large mechanical typical understanding entropy | entropy mechanical typical atoms identify calculations explain explained explanation understanding (0.074) | 4 |
| | break breath continued fast leave objective periods pressed skip slight stating temporary pause | periods temporary pause pressed slight stating fast leave break continued (0.022) | 5 |
| | alive cop met objected promises talk victim violent wanting angry | met talk victim violent promises wanting alive objected cop angry (0.010) | 6 |
| | crash deliver fall figure flash heavy likewise masses pad terrain acceleration | terrain heavy deliver crash likewise figure flash masses fall acceleration (-0.033) | 7 |
| | actors camera combine convenient icons inserting manipulate privileges saving scripted editing | convenient actors camera scripted inserting combine manipulate saving icons editing (-0.141) | 8 |

Table 24: LLaMA-13B, Path Hi-C, Layer 3 Neuron 224, *pos* pool. Websites (#1), mathematics (#2, 3), movement (#5), police-interaction (#6), physics-related (#7) and acting-related (#8).

| Word pool | Word sets | Extracted Topics (NPMI$_W$) | # |
|---|---|---|---|
| access action amity application avery background bail bed belly block bone break brig bsc capped cause cbi charge chase chemist citi class code color con connected connection conning context control custom digital distinct diving dsc dust editing education end eng eps extension fashion fbi fins force forensic fourteenth frames fuse geary gen general genes geology gold hand handful haryana havana hex high inning ips issued jail james jane kane kits lane lens lets level library lieutenant life like line lobe lobed location lone lps macy main management mary master mastered mastering middle midline mining model mold molded molding mounted msc music neural occupied offense office offline outing parking physicist pits play power practice pre primary probe program protection psychology publicity quality rare realistic recording refit resin ret retail retailers room rotation runs scam school schooling sciences scratch scratched scratches screen security sent server service shape size small social space special sport sporting start starts store string structure student students style subsequently suffered support surprise system target targeted targeting test tls toes tooth toothed tops training versus video videos visited walk wall web wine witness witnesses wounded
(486 ommitted words) | bed chemist class dsc education eng geology level life management master mining msc physicist program psychology school schooling sciences social sport student students subsequently training bsc | sciences program education bsc psychology student students msc school master (0.176) | 1 |
| | chase fashion location occupied parking retail retailers space sporting store target macy | retail retailers location space occupied macy fashion parking store target (0.129) | 2 |
| | color diving fins like lobe middle shape size small structure style toes toothed lobed | like toothed size shape lobe color lobed fins middle small (0.120) | 3 |
| | access application extension library security server support system web tls | library access extension support application tls security system web server (0.120) | 4 |
| | action background bail charge connection fbi forensic haryana high ips issued jail jane office probe scam sent special witness witnesses cbi | scam bail charge jail special connection probe fbi sent cbi (0.108) | 5 |
| | avery break capped citi end fourteenth frames lone offense outing play recording rotation start starts versus walk inning | play inning start starts lone walk recording break end offense (0.063) | 6 |
| | block custom gold kits lens model mold molding resin molded | molded lens molding gold kits block custom model resin mold (0.062) | 7 |
| | context handful lps music power rare string video videos eps | music lps string rare context eps handful video videos power (0.034) | 8 |
| | cause digital dust hand pits protection quality scratch scratched screen suffered scratches | quality hand scratch scratches dust cause screen digital protection suffered (0.033) | 9 |
| | belly bone connected distinct fuse neural runs tooth wall midline | fuse bone tooth belly midline neural runs connected wall distinct (0.026) | 10 |
| | con control main mounted practice primary refit room service tops wounded conning | conning mounted wounded tops control room refit primary main service (0.016) | 11 |
| | code force gen general genes lieutenant line pre test ret | pre ret lieutenant test code force gen genes line general (-0.006) | 12 |
| | amity geary havana james kane lane mary surprise visited wine brig | visited james mary brig surprise kane lane wine amity havana (-0.098) | 13 |
| | hex lets mastered mastering offline publicity realistic targeted targeting editing | targeted targeting editing lets hex publicity realistic offline mastered mastering (-0.197) | 14 |

Table 25: LLaMA-13B, Path Lo-C, Layer 36 Neuron 592, *pos* pool. Education (#1), retail (#2), body-parts-related (#3, 6, 10), software (#4), law & order (#5), cricket (#6), sculpting (#7), music videos (#8), surface-damage (#9), and mixed (#11, 12, 13, 14).

| Word pool | Word sets | Extracted Topics (NPMI$_W$) | # |
|---|---|---|---|
| annotations audi authority avoided balls beat book books catch city comments concurrent devices displays dropped dual eventually fill filled finish finished fog functions gates gene generates goals gold google hid institutions internally jurisdiction launched majority micro mouse multiple municipal municipalities orange outer overall overlap patch plants platforms population power presented processor purposes refresh robot rust sant sap second slope smart span species starting surprising systems thick tool transparent trap usage uses variance virtual walls works (91 ommitted words) | concurrent devices displays dual gates generates google internally launched micro mouse platforms power refresh robot sap smart systems usage virtual processor | devices systems usage robot processor displays virtual smart google platforms (0.114) | 1 |
| | audi avoided balls beat catch dropped eventually finish goals gold overall second starting surprising finished | beat goals overall finish finished balls dropped starting second catch (0.083) | 2 |
| | authority city functions institutions jurisdiction majority municipal population purposes sant municipalities | jurisdiction purposes city institutions functions municipal municipalities authority majority population (0.081) | 3 |
| | fill filled fog hid orange outer overlap patch plants rust slope span species transparent trap walls thick | fill filled plants species walls transparent thick outer patch orange (0.072) | 4 |
| | book books comments gene multiple presented tool uses variance works annotations | tool books annotations uses works comments presented gene multiple book (0.031) | 54 |

Table 26: LLaMA-13B, Path Lo-I, Layer 30 Neuron 704, *pos* pool. Internet-of-things (#1), competition (#2), government (#3), nature (#4), and literature (# 5).

| Word pool | Word sets | Extracted Topics (NPMI$_W$) | # |
|---|---|---|---|
| active anonymous aspect catalog comment conservation description dual funds header landscape launch listing literally marker mole museum names pet photos practical profile profiles recommend recommended reliable repository situ society static suffix syntax technical term type verb website websites word (90 ommitted words) | active aspect dual literally marker static suffix syntax term type word verb | syntax suffix aspect term type marker verb literally word dual (0.099) | 1 |
| | anonymous catalog comment header launch listing names photos profile profiles recommend reliable repository technical website websites | profile profiles photos anonymous website websites technical comment names launch (0.076) | 2 |
| | description funds landscape mole museum pet practical recommended situ society conservation | funds society description pet landscape conservation situ practical recommended museum (-0.039) | 3 |

Table 27: LLaMA-7B, Path Lo-C, Layer 30 Neuron 460, *pos* pool. Linguistics (#1) and archival (#2, 3).

| Word pool | Word sets | Extracted Topics (NPMI$_W$) | # |
|---|---|---|---|
| acid announced basically better black blackish blog cease clean comment compact controls cut embedded fins flows forth fraction fresh furnace hours jet kitchen laid load love model modified paired pipe portion posted press product publication random randomly reuse said select short shortly small smart subset sum tail tall think tweet waste (102 ommitted words) | black blackish embedded forth fresh jet modified paired portion short small tail tall fins | black paired tall portion short blackish modified fins tail small (0.089) | 1 |
| | announced blog comment hours love posted press publication shortly think tweet | publication tweet comment announced press love blog think posted shortly (0.069) | 2 |
| | acid compact controls fraction model product random randomly said select sum subset | product fraction said model select compact random sum subset randomly (0.046) | 3 |
| | basically better cease clean cut flows furnace kitchen laid load pipe reuse smart waste | load kitchen waste better pipe clean cut laid furnace flows (0.025) | 4 |

Table 28: LLaMA-7B, Path Lo-C, Layer 2 Neuron 460, *pos* pool. Fish-related (#1), social-media (#2), computation (#3), and plumbing (#4).