# OpenReview forum: "Disentangling Transformer Language Models as Superposed Topic Models"
_EMNLP/2023/Conference — EMNLP 2023 Main_

### Official Review · Reviewer_K1Jc · 2023-07-22

**Soundness:** 4

**Excitement:**

4: Strong: This paper deepens the understanding of some phenomenon or lowers the barriers to an existing research direction.

**Missing References:**

should be "BERTopic"

**Paper Topic And Main Contributions:**

-Disentangling Decoder-Only Transformer-Based Language Models (TLM) to extract coherent topics
- Basically Zero-Shot Topic Modeling on trained large language models

**Questions For The Authors:**

- How does the model compare to the other benchmark models in terms of embedding coherence (see e.g. Octis)
- How does it compare to the other models in terms of topic diversity

- See comment from above

**Reasons To Accept:**

- Novel idea
- Generally well written paper
- Leveraging pre-trained LLMs could highly benefit the area of topic models
- good experimental results

**Reasons To Reject:**

I must admit that I am facing some difficulty in comprehending the concept of "projecting to a corpus" despite carefully reviewing the presented materials. It is possible that my understanding of this particular aspect might be influenced by my own perspective or prior knowledge.

Regarding the methodology and interpretation of the results, it seems that the primary contribution of the LLMs in your algorithm lies in the creation of the word pool. However, creating great NPMI-topics can already be accomplished by solely utilizing word-co-occurrence statistics from the entire corpus and subsequently applying hierarchical clustering to these word-word statistics. I wonder whether the involvement of LLMs may be considered somewhat redundant in this context. I would appreciate it if you could kindly shed some light on whether it is necessary to create the word pools initially or if the graph construction could be achieved without this step.


-Using NPMI as a metric used in creating your topics as well as using it to evaluate your models seems unfair.
-The evaluation is fairly limited. Other metrics should be considered.
-Document-topic matrix not analyzed/result of algorithm.

**Reproducibility:**

2: Would be hard pressed to reproduce the results. The contribution depends on data that are simply not available outside the author's institution or consortium; not enough details are provided.

**Reviewer Confidence:**

2: Willing to defend my evaluation, but it is fairly likely that I missed some details, didn't understand some central points, or can't be sure about the novelty of the work.

**Typos Grammar Style And Presentation Improvements:**

-Sentence starting in line 052 is not a complete sentence.

---

> ### Author Rebuttal · Authors · 2023-08-25
>
> Thank you for the feedback. We will work to improve our communication of the problem and our approach.
>
> We would like to put in a clarification that in the field of topic modeling, the subject of interest is the corpus, models/approaches are developed to analyze the corpus. Whereas in the **mechanistic interpretability** field, the subject of interest is the model (LLM/TLM), and we develop approaches to analyze LLMs to increase our white-box understanding of the inner workings of LLMs. The objective of our work is to interpret and investigate TLMs at the neuron level for explanability. We use the corpus as a tool to constrain our scope of investigation within the TLM. Essentially, our work applies topic modeling evaluation methodology to the mechanistic interpretability field.
>
> There are many challenges in the LLM mechanistic interpretability field, and our work attempts to reduce the barriers (some listed here for convenience):
>
> (1) Current neuron-level interpretation of LLMs is limited. Usually, one neuron is mapped to a concept (represented as a topic) on the token space, and interpreting each neuron for human understanding in an LLM is challenging. We show an empirical approach to map one neuron to many human-coherent concepts.
>
> (2) Most LLM interpretability work is done on GPT2, due to the challenges of working with LLaMA's tokens. We propose a viable option on how to advance the field on LLaMA's token space.
>
> (3) Many existing interpretation studies are not quantifiable at a large scale and require similarly large user studies, which only well-funded institutes can conduct at scale. We apply existing topic modeling evaluation methodology to show that automation to measure interpretability can be achieved, lowering barriers to entry.
>
> **On the involvement of LLM:**
> We have to involve LLM as we are trying to investigate LLM.
>
> **On creating word pools:**
> As our subject of interest is the LLM, the word pools are created from the logit distribution of the individual neurons. The corpus is a tool to restrict the scope of the investigation. By projecting to the corpus, we are analysing if the behaviour of the neuron of the LLM emulate similar information found in the corpus.
>
> **On hierarchical clustering:**
> We are aware of the methods. We considered it and decided against it due to three reasons:
>
> i. It adds another layer of complexity to the approach, with additional consideration to which algorithm, distance function, and features to use.
>
> ii. The size of word pools will vary as LLaMA's word pools do not have a fixed size.
>
> iii. The size and number of word sets will also vary between word pools. The coherent topics extracted might not reflect the word pool due to the presence of outliers in the clustering process.
>
> **On topic diversity:**
> For section 7, we evaluated Topic Diversity as TU, the proportion of unique words in the set of Topics, and it is in the paper. Our results show topic sets extracted from LLM are comparable or better.
>
> For sections 6 and 8, since the number of topics generated far exceeds the vocabulary size of Wiki corpus, evaluating TU is not necessary. With regards to possible duplicate topics, we remove topics with words that occur very frequently across the entire set of topics. Otherwise, it is acceptable when different neurons in the LLM express similar topics.
>
> **On the fairness of using NPMI:**
> We are evaluating NPMI as the tool to extract human-coherent interpretations from LLMs instead of using NPMI to evaluate our approach.
>
> Our reasons for using NPMI is rooted in Wiki's NPMI$_W$'s correlation with human judgment (line 140-150), and we wish to analyze the LLM in a human-coherent manner. For sections 6 and 8, we use NPMI$_W$ to extract human-coherent topic representations from LLMs, which we compare to extracted topic representations from a random distribution as the baseline. Our results show that the topics from LLMs are better, suggesting that the interpretation approach is meaningful and not due to chance.
>
> For section 7, NPMI$_C$ differs from NPMI$_W$, as NPMI$_C$'s correlation with human judgment is unknown. ProdLDA and CTM are augmented with an NPMI$_C$ optimizing variant.
>
> **Zero-shot Topic Modeling (Section 7):**
> Zero-shot Topic Modeling is not the goal of this paper. We use this task to evaluate whether the superposed topics derived from the neurons are reflective of their extracted superposed topics, and thus meaningful. Given a corpus, we expect it to activate neurons with similar extracted topics. To validate this hypothesis, we must derive an equivalent or better set of coherent topic representations from the activated LLM neurons. Otherwise, there will be a contradiction if the top activated neurons are unable to produce a set of topics reflective of the corpus.
>
> **On embedding coherence:**
> It is beyond the scope of our work. Our work focuses on interpreting LLMs at the neuron level and their corresponding final logit distribution. As mentioned in our related works section, there are other research works on LLMs interpretation in the embedding space.
>
>
> Our goal is to extend the currently accepted methodology of evaluating the interpretability of Topic Models to evaluate the interpretability of LLM at the neuron level, lowering barriers to the field of mechanistic interpretability. We hope our clarification and answers have shed additional light and show that our work is not just a zero-shot topic modeling paper.

---

### Official Review · Reviewer_AWCb · 2023-08-03

**Soundness:** 4

**Excitement:**

4: Strong: This paper deepens the understanding of some phenomenon or lowers the barriers to an existing research direction.

**Paper Topic And Main Contributions:**

This paper proposed a weight based, model agnostic and corpus-agnostic method to search and disentangle decoder based Transformer Language Models, the method convert the topic disentanglement problem into a graph optimisation task. Experiments on GPT2 and Llama shows the effectiveness of this method.

**Questions For The Authors:**

Question A: Is your method available for large language models such as GPT3 ?

**Reasons To Accept:**

* The interpretation of TLM as superposed NTM is novel.
* The three step disentanglement process is clear: the project step is used to get corpus statistics, the shortlisting step used some heuristics to create smaller graphs and the exact-solving step locate the subgraphs and return coherent topics.
* Experiments using widely used language models such as GPT2 and Llama are impressive.

**Reasons To Reject:**

* Not sure whether this method is scalable, as I see the old small data sets for topic modelling (e.g. 20 Newsgroup), it would be better to analyse the computation complexity when the word graph is huge.

**Reproducibility:**

3: Could reproduce the results with some difficulty. The settings of parameters are underspecified or subjectively determined; the training/evaluation data are not widely available.

**Reviewer Confidence:**

3: Pretty sure, but there's a chance I missed something. Although I have a good feel for this area in general, I did not carefully check the paper's details, e.g., the math, experimental design, or novelty.

---

> ### Author Rebuttal · Authors · 2023-08-25
>
> R1: In Sections 6 and 8, our Wiki corpus has a billion tokens (5M documents, 40K vocabulary). Our Python code produces the word graphs for our Wiki corpus in 6 hours using CPU AMD EPYC 7502 @ 2.50GHz utilizing 40 cores. A similar test on a larger corpus with 1.5 billion tokens (1M documents, 40K vocabulary) produces its word graph in 7 hours. The theoretical complexity is O(Num. docs x (vocab. in doc.)$^2$). We will release our implementation code.
>
> Wiki is the largest corpus that have been used for coherence measure. No larger corpus has been benchmarked and shown better correlation to human judgment.
>
> In Section 7, the purpose of the zero-shot topic modeling task is to verify whether we can obtain a set of relevant coherent topics from TLM's top activated neurons when we use the corpus as the input. This further validates the link between the neuron and its extracted superposed topics. We opted for smaller datasets for this validation, as the corpora are well understood. In practice, since this is a validation step, we do not need to pass the corpus into the TLM to identify the top activated neurons for topic extraction.
>
> QA: Our approach should work on any TLM, provided the user can access the model's weights. Producing the logit distribution of each neuron has a similar complexity to next token generation. We expect GPT3 to have many more neurons, requiring more resources to evaluate the interpretability of the model. GPT3 is still closed-source at the time of writing, if it is publicly available, a similar analysis could be conducted.

---

### Official Review · Reviewer_Y4Bt · 2023-08-03

**Soundness:** 4

**Excitement:**

4: Strong: This paper deepens the understanding of some phenomenon or lowers the barriers to an existing research direction.

**Paper Topic And Main Contributions:**

This paper offers a novel perspective to derive meaningful topics from transformer language models.
This paper proposes a weight-based, model-agnostic and corpus-agnostic approach to search and disentangle decoder-only TLM, mapping individual neurons to multiple coherent topics.

**Questions For The Authors:**

A. The zero shot topic modeling method proposed in the paper is attractive. But how does the projection mechanism work when some words in the corpus cannot match the tokens of the large language model?
B. What are the main advantages of this method compared with traditional neural topic models which leverage the embedding of large language models for topic modeling,

**Reasons To Accept:**

1. The method of deriving topics looks novel to me.
2. The experimental section of this paper is remarkably comprehensive.

**Reasons To Reject:**

1. This paper may be hard to understand by some readers who are not familiar with disentangling superpositions. More preliminary introductions will make the paper easier to follow.

**Reproducibility:**

4: Could mostly reproduce the results, but there may be some variation because of sample variance or minor variations in their interpretation of the protocol or method.

**Reviewer Confidence:**

3: Pretty sure, but there's a chance I missed something. Although I have a good feel for this area in general, I did not carefully check the paper's details, e.g., the math, experimental design, or novelty.

---

> ### Author Rebuttal · Authors · 2023-08-25
>
> R1: We will use the extra page to improve on this.
>
> QA: In Section 8, we demonstrate this use case. Roughly 90% of LLaMA's 32K tokens are sub-words, and many words from our Wiki corpus do not exist in LLaMA's token space. We empirically demonstrate that we can treat words as a "topic" representation of tokens (sub-words), forming word pools to extract coherent topics.
>
> QB: Traditional neural topic models (with LLM embeddings) "explain" corpus, whereas our approach seeks to interpret and examine LLMs in a white-box setting. We are unaware of any neural topic models for extremely large corpus since the resources required are logically better used on general-purpose LLMs. Our proposed approach examines LLM's knowledge, which is pre-trained on these billions (or trillion) token corpus, and presents its analysis in a human-coherent manner. In doing so, we are applying topic modeling evaluation methodology in the field of mechanistic interpretability.

---

### Meta-Review · Area_Chair_TH1o · 2023-09-16

**Recommendation:** 4

**Metareview:**

The paper proposes a novel approach to topic modeling using decoder-only Transformer-Based Language Models (TLMs). The proposed approach is clear and easy to follow. All reviewers agree on the novelty of the paper's approach which executes the idea well and evaluates it thoroughly, including analysis of quantitative evaluation and extending to LLaMA. Experiments using widely used language models such as GPT2 and Llama are impressive and remarkably comprehensive. From the experimental results, we can find that leveraging pre-trained TLMs could highly benefit the area of topic models.

---

### Decision · Program_Chairs · 2023-10-07

**Decision:**

Accept-Main

**Comment:**

The paper proposes a novel approach to topic modeling using decoder-only Transformer-Based Language Models (TLMs). The proposed approach is clear and easy to follow. All reviewers agree on the novelty of the paper's approach which executes the idea well and evaluates it thoroughly, including analysis of quantitative evaluation and extending to LLaMA. Experiments using widely used language models such as GPT2 and Llama are impressive and remarkably comprehensive. From the experimental results, we can find that leveraging pre-trained TLMs could highly benefit the area of topic models.